# Differentiable Scaffolding Tree for Molecular Optimization

**Tianfan Fu**[1*]**, Wenhao Gao**[2*]**, Cao Xiao**[3]**, Jacob Yasonik**[2]**, Connor W. Coley**[2] **& Jimeng Sun**[4]
[1]Georgia Institute of Technology, [2]Massachusetts Institute of Technology
[3]Amplitude. [4]University of Illinois at Urbana-Champaign. [*]Equal Contribution

`tfu42@gatech.edu`   `{whgao,jyasonik,ccoley}@mit.edu`
`{danicaxiao,jimeng.sun}@gmail.com`

## Abstract

The structural design of functional molecules, also called molecular optimization, is an essential chemical science and engineering task with important applications, such as drug discovery. Deep generative models and combinatorial optimization methods achieve initial success but still struggle with directly modeling discrete chemical structures and often heavily rely on brute-force enumeration. The challenge comes from the discrete and non-differentiable nature of molecule structures. To address this, we propose differentiable scaffolding tree (`DST`) that utilizes a learned knowledge network to convert discrete chemical structures to locally differentiable ones. `DST` enables a gradient-based optimization on a chemical graph structure by back-propagating the derivatives from the target properties through a graph neural network (GNN). Our empirical studies show the gradient-based molecular optimizations are both effective and sample efficient (in terms of oracle calling number). Furthermore, the learned graph parameters can also provide an explanation that helps domain experts understand the model output. The code repository (including processed data, trained model, demonstration, molecules with the highest property) is available at `https://github.com/futianfan/DST`.

## 1 Introduction

The structural design of new functional molecules, also called molecular optimization, is the key to many scientific and engineering challenges, such as finding energy storage materials (Hachmann et al., 2011; Janet et al., 2020), small molecule pharmaceutics (Kuntz, 1992; Zhavoronkov et al., 2019), and environment-friendly material (Zimmerman et al., 2020). The objective is to identify novel molecular structures with desirable chemical or physical properties (Gómez-Bombarelli et al., 2018; Dai et al., 2018; Jin et al., 2018; You et al., 2018; Jin et al., 2019; Shi et al., 2020; Zhou et al., 2019; Fu et al., 2020; Jin et al., 2020; Zang & Wang, 2020; Xie et al., 2021). Recent advances in deep generative models (DGM) allow learning the distribution of molecules and optimizing the latent embedding vectors of molecules. Models in this category are exemplified by the variational autoencoder (VAE) (Gómez-Bombarelli et al., 2018; Dai et al., 2018; Jin et al., 2018; 2020) and generative adversarial network (GAN) (De Cao & Kipf, 2018). On the other hand, because of the discrete nature of the enormous chemical space, applying combinatorial optimization algorithms with some structure enumeration has been the predominant approach (You et al., 2018; Jensen, 2019; Zhou et al., 2019; Nigam et al., 2020; Xie et al., 2021). Deep learning models have also been used to guide these combinatorial optimization algorithms. For example, You et al. (2018); Zhou et al. (2019); Jin et al. (2020); Gottipati et al. (2020) tried to solve the problem with deep reinforcement learning; Nigam et al. (2020) enhanced a genetic algorithm with a neural network as a discriminator; Xie et al. (2021); Fu et al. (2021) approached the problem with Markov Chain Monte Carlo (MCMC) to explore the target distribution guided by graph neural networks.

Despite the initial success of these previous attempts, the following limitations remain: (1) deep generative models optimize the molecular structures in a learned latent space, which requires the latent space to be smooth and discriminative. Training such models need carefully designed networks and well-distributed datasets. (2) most combinatorial optimization algorithms, featured by evolutionary

learning methods (Nigam et al., 2020; Jensen, 2019; Xie et al., 2021; Fu et al., 2021), exhibit random-walk behavior, and leverage trial-and-error strategies to explore the discrete chemical space. The recent deep reinforcement learning methods (You et al., 2018; Zhou et al., 2019; Jin et al., 2020; Gottipati et al., 2020) aim to remove random-walk search using a deep neural network to guide the searching. It is challenging to design the effective reward function into the objective (Jin et al., 2020). (3) Most existing methods require a significant number of oracle calls (a property evaluator; see Def. 1) to proceed with an efficient search. However, realistic oracle functions, evaluating with either experiments or high-fidelity computational simulations, are usually expensive. Examples include using biological assays to determine the potency of drug candidates (Wang et al., 2017), or conducting electronic structure calculation to determine photoelectric properties (Long et al., 2011).

We propose differentiable scaffolding tree (`DST`) to address these challenges, where we define a differentiable scaffolding tree for molecular structure and utilize a trained GNN to obtain the local derivative that enables continuous optimization. The main contributions are summarized as:

• We propose the differentiable scaffolding tree to define a local derivative of a chemical graph. This concept enables a gradient-based optimization of a discrete graph structure.

• We present a general molecular optimization strategy utilizing the local derivative defined by the differentiable scaffolding tree. This strategy leverages the property landscape's geometric structure and suppresses the random-walk behavior, exploring the chemical space more efficiently. We also incorporate a determinantal point process (DPP) selection strategy to enhance the diversity of generated molecules.

• We demonstrate encouraging preliminary results on *de novo* molecular optimization with multiple computational objective functions. The local derivative shows consistency with chemical intuition, providing **interpretability** of the chemical structure-property relationship. Our method also requires less **oracle calls**, maintaining good performance in limited oracle settings.

## 2 RELATED WORK

Existing molecular optimization methods can mainly be categorized as deep generative models and combinatorial optimization methods.

**Deep generative models** model a distribution of general molecular structure with a deep network model so that one can generate molecules by sampling from the learned distribution. Typical algorithms include variational autoencoder (VAE), generative adversarial network (GAN), energy-based models, flow-based model (Gómez-Bombarelli et al., 2018; Jin et al., 2018; De Cao & Kipf, 2018; Segler et al., 2018; Jin et al., 2019; Honda et al., 2019; Madhawa et al., 2019; Shi et al., 2020; Jin et al., 2020; Zang & Wang, 2020; Kotsias et al., 2020; Chen et al., 2021; Fu et al., 2020; Liu et al., 2021; Bagal et al., 2021). Shen et al. (2021) also leverages inverse learning based on SELFIES representation. However, its performance is not satisfactory, primarily due to the failure of training an adequate surrogate oracle. In addition, DGMs can leverage Bayesian optimization in latent spaces to optimize latent vectors and reconstruct to obtain the optimized molecules (Jin et al., 2018). However, such approaches usually require a smooth and discriminative latent space and thus an elaborate network architecture design and well-distributed data set. Also, as they learn the reference data distribution, their ability to explore diverse chemical space is relatively limited, evidenced by the recent molecular optimization benchmarks (Brown et al., 2019; Huang et al., 2021).

**Combinatorial optimization methods** mainly include deep reinforcement learning (DRL) (You et al., 2018; Zhou et al., 2019; Jin et al., 2020; Gottipati et al., 2020) and evolutionary learning methods (Nigam et al., 2020; Jensen, 2019; Xie et al., 2021; Fu et al., 2021). They both formulate molecule optimization as a discrete optimization task. Specifically, they modify molecule substructures (or tokens in a string representation (Weininger, 1988)) locally, with an oracle score or a policy/value network to tell if they keep it or not. Due to the discrete nature of the formulation, most of them conduct an undirected search (random-walk behavior), while some recent ones like reinforcement learning try to guide the searching with a deep neural network, aiming to rid the random-walk nature. However, it is challenging to incorporate the learning objective target into the guided search. Those algorithms still require massive numbers of oracle calls, which is computationally inefficient during the inference time (Korovina et al., 2020). Our method, `DST`, falls into this category, explicitly leverages the objective function landscape, and conducts an efficient goal-oriented search. Instead

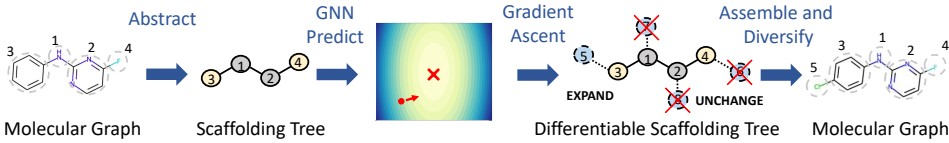

Figure 1: Illustration of the overall approach: During inference, we construct the corresponding scaffolding tree and differentiable scaffolding tree (`DST`) for each molecule. We optimize each `DST` along its gradient back-propagated from the GNN and sample scaffolding trees from the optimized `DST`. After that, we assemble trees into molecules and diversify them for the next iteration.

of operating on molecular substructure or tokens, we define the search space as a set of binary and multinomial variables to indicate the existence and identity of nodes, respectively, and make it locally differentiable with a learned GNN as a surrogate of the oracle. This problem formulation can find its root in conventional computer-aided molecular design algorithms with branch-and-bound algorithms as solutions (Sinha et al., 1999; Sahinidis & Tawarmalani, 2000).

## 3 METHOD

We first introduce the formulation of molecular optimization and *differentiable scaffolding tree* (`DST`) in Section 3.1, illustrate the pipeline in Figure 1, then describe the key steps following the order:

- **Oracle GNN construction:** We leverage GNNs to imitate property oracles, which are targets of molecular optimization (Section 3.2). Oracle GNN is trained once and for all. The training is separately from optimizing `DST` below.

- **Optimizing differentiable scaffolding tree:** We formulate the discrete molecule optimization into *a locally differentiable* problem with a differentiable scaffolding tree (`DST`). Then a `DST` can be optimized by the gradient back-propagated from oracle GNN (Section 3.3).

- **Molecule Diversification** After that, we describe how we design a *determinantal point process (DPP)* based method to output diverse molecules for iterative learning (Section 3.4).

### 3.1 PROBLEM FORMULATION AND NOTATIONS

**3.1.1 Molecular optimization problem** Oracles are the objective functions for molecular optimization problems, e.g., QED quantifying a molecule's drug-likeness (Bickerton et al., 2012).

**Definition 1** (Oracle $\mathcal{O}$). *Oracle $\mathcal{O}$ is a black-box function that evaluates certain chemical or biological properties of a molecule $X$ and returns the ground truth property $\mathcal{O}(X)$.*

In realistic discovery settings, the oracle acquisition cost is usually not negligible. Suppose we want to optimize $P$ molecular properties specified by oracle $\mathcal{O}_1, \cdots, \mathcal{O}_P$, we can formulate a multi-objective molecule optimization problem through scalarization as represented in Eq. (1),

$$\arg\max_{X \in \mathcal{Q}} \ F(X; \mathcal{O}_1, \mathcal{O}_2, \cdots, \mathcal{O}_P) = f(\mathcal{O}_1(X), \cdots, \mathcal{O}_P(X)), \tag{1}$$

where $X$ is a molecule, $\mathcal{Q}$ denotes the set of valid molecules; $f$ is the composite objective combining all the oracle scores, e.g., the mean value of $P$ oracle scores.

**3.1.2 Scaffolding Tree** The basic mathematical description of a molecule is a molecular graph, which contains atoms as nodes and chemical bonds as edges. However, molecular graphs are not easy to generate explicitly as graphs due to the presence of rings, relatively large size, and chemical validity constraints. For ease of computation, we convert a molecular graph to a scaffolding tree as a higher-level representation, a tree of substructures, following Jin et al. (2018; 2019).

**Definition 2** (**Substructure**). *Substructures can be either an atom or a single ring. The substructure set is denoted $\mathcal{S}$ (vocabulary set), covering frequent atoms and single rings in drug-like molecules.*

**Definition 3** (**Scaffolding Tree $\mathcal{T}$**). *A scaffolding tree, $\mathcal{T}_X$, is a spanning tree whose nodes are substructures. It is a higher-level representation of molecular graph $X$.*

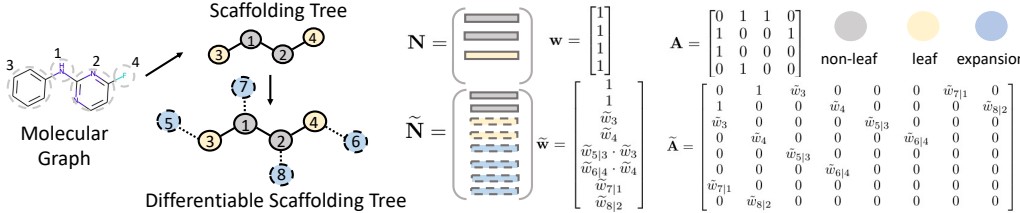

Figure 2: Example of differentiable scaffolding tree. We show non-leaf nodes (grey), leaf nodes (yellow), expansion nodes (blue). The dashed nodes and edges are learnable, corresponding to nodes' identity and existence. $\widetilde{\mathbf{w}}$ and $\widetilde{\mathbf{A}}$ share the learnable parameters $\{\widehat{\mathbf{w}}_3, \widehat{\mathbf{w}}_4, \widehat{\mathbf{w}}_{5|3}, \widehat{\mathbf{w}}_{6|4}, \widehat{\mathbf{w}}_{7|1}, \widehat{\mathbf{w}}_{8|2}\}$.

$\mathcal{T}_X$ is represented by (i) node indicator matrix, (ii) adjacency matrix, and (iii) node weight vector. We distinguish leaf and non-leaf nodes in $\mathcal{T}_X$. Among the $K$ [1] nodes in $\mathcal{T}_X$, there are $K_{\text{leaf}}$ leaf nodes (nodes connecting to only one edge) and $K - K_{\text{leaf}}$ non-leaf nodes (otherwise). The sets of leaf nodes and non-leaf nodes are denoted $\mathcal{V}_{\text{leaf}}$ and $\mathcal{V}_{\text{nonleaf}}$ correspondingly.

**Definition 4.** *Node indicator matrix* $\mathbf{N}$ *is decomposed as* $\mathbf{N} = \begin{pmatrix} \mathbf{N}_{nonleaf} \\ \mathbf{N}_{leaf} \end{pmatrix} \in \{0,1\}^{K \times |\mathcal{S}|}$, *where*

$\mathbf{N}_{nonleaf} \in \{0,1\}^{(K-K_{leaf}) \times |\mathcal{S}|}$ *corresponds to non-leaf nodes while* $\mathbf{N}_{leaf} \in \{0,1\}^{K_{leaf} \times |\mathcal{S}|}$ *corresponds to leaf nodes. Each row of* $\mathbf{N}$ *is a one-hot vector, indicating which substructure the node belongs to.*

**Definition 5.** *Adjacency matrix is denoted* $\mathbf{A} \in \{0,1\}^{K \times K}$. $A_{ij} = 1$ *indicates the $i$-th node and $j$-th node are connected while 0 indicates unconnected.*

**Definition 6.** *Node weight vector,* $\mathbf{w} = [1, \cdots, 1]^\top \in \mathbb{R}^K$, *indicates the $K$ nodes in scaffolding tree are equally weighted.*

**3.1.3 Differentiable scaffolding tree** Similar to a scaffolding tree, a differentiable scaffolding tree (DST) also contains (i) node indicator matrix, (ii) adjacency matrix, and (iii) node weight vector, but with additional expansion nodes. Specifically, while inheriting leaf node set $\mathcal{V}_{\text{leaf}}$ and non-leaf node set $\mathcal{V}_{\text{nonleaf}}$ from the original scaffolding tree, we add expansion nodes and form *expansion node set*, $\mathcal{V}_{\text{expand}} = \{u_v | v \in \mathcal{V}_{\text{leaf}} \cup \mathcal{V}_{\text{nonleaf}}\}, |\mathcal{V}_{\text{expand}}| = K_{\text{expand}} = K$, where $u_v$ is connected to $v$ in the original scaffolding tree. We also define *differentiable edge set*, $\Lambda = \{(v, v') \mid v \in \mathcal{V}_{\text{leaf}} \text{ OR } v' \in \mathcal{V}_{\text{expand}}; v, v' \text{ are connected}\}$ to incorporate all the edges involving leaf-nonleaf node and leaf/nonleaf-expansion node connections. To make it locally differentiable, we modify the tree parameters from two aspects: (A) *node identity* and (B) *node existence*. Figure 2 shows an example to illustrate DST.

(A) We enable optimization on *node identity* by allowing the corresponding node indicator matrix learnable:

**Definition 7.** *Differentiable node indicator matrix* $\widetilde{\mathbf{N}}$ *takes the form:*

$$\widetilde{\mathbf{N}} = \begin{pmatrix} \widetilde{\mathbf{N}}_{nonleaf} \\ \widetilde{\mathbf{N}}_{leaf} \\ \widetilde{\mathbf{N}}_{expand} \end{pmatrix} \in \mathbb{R}_+^{(K+K_{expand}) \times |\mathcal{S}|}, \quad \sum_{j=1}^{|\mathcal{S}|} \widetilde{\mathbf{N}}_{ij} = 1, \quad K = K_{expand}. \tag{2}$$

$\widetilde{\mathbf{N}}_{nonleaf} = \mathbf{N}_{nonleaf} \in \{0,1\}^{(K-K_{leaf}) \times |\mathcal{S}|}$ *are fixed, equal to the part in the original scaffolding tree, each row is a one-hot vector, indicating that we fix all the non-leaf nodes. In contrast, both* $\widetilde{\mathbf{N}}_{expand}$ *and* $\widetilde{\mathbf{N}}_{leaf}$ *are learnable, we use softmax activation to implicitly encode the constraint* $\sum_j \widetilde{\mathbf{N}}_{ij} = 1$ *i.e.,* $\widetilde{\mathbf{N}}_{ij} = \frac{\exp(\widehat{\mathbf{N}_{ij}})}{\sum_{j'=1}^{|\mathcal{S}|} \exp(\widehat{\mathbf{N}_{i,j'}})}$, $\widehat{\mathbf{N}}$ *are the parameters to learn. This constraint guarantee that each row of* $\widetilde{\mathbf{N}}$ *is a valid substructures' distribution.*

(B) We enable optimization on *node existence* by assigning learnable weights for the leaf and expansion nodes, constructing adjacency matrix and node weight vector based on those values:

---

[1] $K$ depends on molecular graph. During optimization (Section 3.3 and 3.4), after molecular structure changes, $K$ is updated.

**Definition 8.** *Differentiable adjacency matrix $\widetilde{\mathbf{A}} \in \mathbb{R}^{(K+K_{expand}) \times (K+K_{expand})}$ takes the form:*

$$\widetilde{\mathbf{A}}_{ij} = \widetilde{\mathbf{A}}_{ji} = \begin{cases} 1/0, & (i,j) \notin \Lambda \ 0\text{:}disconnected, \ 1\text{:}connected \\ \sigma(\widehat{\mathbf{w}}_i), & (i,j) \in \Lambda, i \in \mathcal{V}_{leaf}, \ j \in \mathcal{V}_{nonleaf} \\ \sigma(\widehat{\mathbf{w}}_{i|j}), & (i,j) \in \Lambda, i \in \mathcal{V}_{expand}, \ j \in \mathcal{V}_{leaf} \cup \mathcal{V}_{nonleaf} \end{cases} \tag{3}$$

*where $\Lambda$ is the differentiable edge set defined above, Sigmoid function $\sigma(\cdot)$ imposes the constraint $0 \leq \widetilde{\mathbf{A}}_{ij} \leq 1$. $\widehat{\mathbf{w}} \in \mathbb{R}^{K_{leaf}+K_{expand}}$ are the parameters, each leaf node and expansion node has one learnable parameter. For connected $i$ and $j$, when $i \in \mathcal{V}_{leaf}$, $j \in \mathcal{V}_{nonleaf}$, $\widetilde{\mathbf{A}}_{ij} = \sigma(\widehat{\mathbf{w}}_i)$ measures the existence probability of leaf node $i$; when $i \in \mathcal{V}_{expand}, j \in \mathcal{V}_{leaf} \cup \mathcal{V}_{nonleaf}$, $\widetilde{\mathbf{A}}_{ij} = \sigma(\widehat{\mathbf{w}}_{i|j})$ measures the conditional probability of the existence of expand node $i$ given the original node $j$. When $j$ is a leaf node, it naturally embeds the inheritance relationship between the leaf node and the corresponding expansion node.*

**Definition 9.** *Differentiable node weight vector $\widetilde{\mathbf{w}} \in \mathbb{R}^{K+K_{expand}}$ takes the form:*

$$\widetilde{\mathbf{w}}_i = \begin{cases} 1, & i \in \mathcal{V}_{nonleaf} \\ \sigma(\widehat{\mathbf{w}}_i), & i \in \mathcal{V}_{leaf} \\ \sigma(\widehat{\mathbf{w}}_{i|j})\sigma(\widehat{\mathbf{w}}_j), & i \in \mathcal{V}_{expand}, j \in \mathcal{V}_{leaf}, \quad (i,j) \in \Lambda, \\ \sigma(\widehat{\mathbf{w}}_{i|j})\widetilde{\mathbf{w}}_j = \sigma(\widehat{\mathbf{w}}_{i|j}), & i \in \mathcal{V}_{expand}, j \in \mathcal{V}_{nonleaf}, \quad (i,j) \in \Lambda, \end{cases} \tag{4}$$

*where all the weights range from 0 to 1. The weight of the expansion node connecting to the leaf node relies on the weight of the corresponding leaf node. $\widetilde{\mathbf{w}}$ and $\widetilde{\mathbf{A}}$ (Def. 8) shares the learnable parameter $\widehat{\mathbf{w}}$. Figure 2 shows an example to illustrate DST.*

## 3.2 TRAINING ORACLE GRAPH NEURAL NETWORK

This section constructs a differentiable surrogate model to capture the knowledge from any oracle function. We choose graph neural network architecture for its state-of-the-art performance in modeling structure-property relationships. In particular, we imitate the objective function $F$ with GNN:

$$\widehat{y} = \text{GNN}(X; \Theta) \approx F(X; \mathcal{O}_1, \mathcal{O}_2, \cdots, \mathcal{O}_P) = y, \tag{5}$$

where $\Theta$ represents the GNN's parameters. Concretely, we use a graph convolutional network (GCN) (Kipf & Welling, 2016). Other GNN variants, such as Graph Attention Network (GAT) (Veličković et al., 2017), Graph Isomorphism Network (GIN) (Xu et al., 2018), can also be used in our setting. The initial node embeddings $\mathbf{H}^{(0)} = \mathbf{NE} \in \mathbb{R}^{K \times d}$ stacks basic embeddings of all the nodes in the scaffolding tree, $d$ is the GCN hidden dimension, $\mathbf{N}$ is the node indicator matrix (Def. 4). $\mathbf{E} \in \mathbb{R}^{|\mathcal{S}| \times d}$ is the embedding matrix of all the substructures in a vocabulary set $\mathcal{S}$, and is randomly initialized. The updating rule of GCN for the $l$-th layer is

$$\mathbf{H}^{(l)} = \text{RELU}\big(\mathbf{B}^{(l)} + \mathbf{A}(\mathbf{H}^{(l-1)}\mathbf{U}^{(l)})\big), \quad l = 1, \cdots, L, \tag{6}$$

where $L$ is GCN's depth, $\mathbf{A}$ is the adjacency matrix (Def. 5), $\mathbf{H}^{(l)} \in \mathbb{R}^{K \times d}$ is the nodes' embedding of layer $l$, $\mathbf{B}^{(l)} = [\mathbf{b}^{(l)}, \mathbf{b}^{(l)}, \cdots, \mathbf{b}^{(l)}]^{\top} \in \mathbb{R}^{K \times d}/\mathbf{U}^{(l)} \in \mathbb{R}^{d \times d}$ are bias/weight parameters.

We generalize the GNN from a discrete scaffolding tree to a differentiable one. Based on learnable weights for each node, we leverage the weighted average as the readout function of the last layer's ($L$-th) node embeddings, followed by multilayer perceptron (MLP) to yield the prediction $\widehat{y}$, i.e., $\widehat{y} = \text{MLP}\big(\frac{1}{\sum_{k=1}^{K} w_k} \sum_{k=1}^{K} w_k H_k^{(L)}\big)$, in discrete scaffolding tree, weights for all the nodes are equal to 1, $H_k^{(L)}$ is the $k$-th row of $H^{(L)}$. In sum, the prediction can be written as

$$\widehat{y} = \text{GNN}(X; \Theta) = \text{GNN}(\mathcal{T}_X = \{\mathbf{N}, \mathbf{A}, \mathbf{w}\}; \Theta), \quad X \in \mathcal{Q} \tag{7}$$

where $\Theta = \{\mathbf{E}\} \cup \{\mathbf{B}^{(l)}, \mathbf{U}^{(l)}\}_{l=1}^{L}$ are the GNN's parameters. We train the GNN by minimizing the discrepancy between GNN prediction $\widehat{y}$ and the ground truth $y$.

$$\Theta_* = \arg\min_{\Theta} \sum_{(X,y) \in \mathcal{D}} \mathcal{L}\big(y = F(X; \mathcal{O}_1, \mathcal{O}_2, \cdots, \mathcal{O}_P), \ \widehat{y} = \text{GNN}(X; \Theta)\big), \tag{8}$$

where $\mathcal{L}$ is loss function, e.g., mean squared error; $\mathcal{D}$ is the training set. After training, we have GNN parameterized by $\Theta_*$ to approximate the black-box objective function $F$ (Eq. 1). Worth mentioning that Oracle GNN is trained once and for all. The training is separately from optimizing DST below.

### 3.3 OPTIMIZING DIFFERENTIABLE SCAFFOLDING TREE

**Overview** With a little abuse of notations, via introducing DST, we approximate molecule optimization as a *locally differentiable* problem

$$\underbrace{X = \arg\max_{X \in \mathcal{Q}} F(X)}_{\text{(I) structured combinatorial optimization}} \approx \underbrace{X^{(t+1)} = \arg\max_{X \in \mathcal{N}(X^{(t)})} F(X)}_{\text{(II) iterative local discrete search}}$$

$$\approx \underbrace{\mathcal{T}_{X^{(t+1)}} = \arg\max_{X \in \mathcal{N}(X^{(t)})} \text{GNN}(\mathcal{T}_X = \{\widetilde{\mathbf{N}}_{X^{(t)}}, \widetilde{\mathbf{A}}_{X^{(t)}}, \widetilde{\mathbf{w}}_{X^{(t)}}\}; \Theta_*),}_{\text{(III) local differentiable optimization}} \quad (9)$$

where $X^{(t)}$ is the molecule at $t$-th iteration, $\mathcal{N}(X^{(t)}) \subseteq \mathcal{Q}$ is the neighborhood set of $X^{(t)}$ (Def. 10). Next, we explain the intuition behind these approximation steps. Molecular optimization is generally a discrete optimization task, which is prohibitively expensive due to exhaustive search. The first approximation is to formulate the problem as an iterative local discrete search via introducing a neighborhood molecule set $\mathcal{N}(X^{(t)})$. Second, to enable differentiable learning, we use GNN to imitate black-box objective $F$ (Section 3.2) and further reformulated it into *a local differentiable optimization* problem. Then we can optimize DST ($\mathcal{T}_X = \{\widetilde{\mathbf{N}}_{X^{(t)}}, \widetilde{\mathbf{A}}_{X^{(t)}}, \widetilde{\mathbf{w}}_{X^{(t)}}\}$) in a continuous domain for $\mathcal{N}(X^{(t)})$ using gradient-based optimization method.

**3.3.1 Local Editing Operations** For a leaf node $v$ in the scaffolding tree, we can perform three editing operations, (1) **SHRINK**: delete node $v$; (2) **REPLACE**: replace a new substructure over $v$; (3) **EXPAND**: add a new node $u_v$ that connects to node $v$. For a nonleaf node $v$, we support (1) **EXPAND**: add a new node $u_v$ connecting to $v$; (2) **do nothing**. If we EXPAND and REPLACE, the new substructures are sampled from the vocabulary $\mathcal{S}$. We define molecule neighborhood set:

**Definition 10** (Neighborhood set). *Neighborhood set of molecule $X$, denoted $\mathcal{N}(X)$, is the set of all the possible molecules obtained by imposing one local editing operation to scaffolding tree $\mathcal{T}_X$ and assembling the edited trees into molecules.*

**3.3.2 Optimizing DST** Then within the domain of neighborhood molecule set $\mathcal{N}(X)$, the objective function can be represented as a differentiable function of X's DST ($\widetilde{\mathbf{N}}_X, \widetilde{\mathbf{A}}_X, \widetilde{\mathbf{w}}_X$). We address the following optimization problem to get the best scaffolding tree within $\mathcal{N}(X)$,

$$\widetilde{\mathbf{N}}_*, \widetilde{\mathbf{A}}_*, \widetilde{\mathbf{w}}_* = \arg\max_{\{\widetilde{\mathbf{N}}_X, \widetilde{\mathbf{A}}_X, \widetilde{\mathbf{w}}_X\}} \text{GNN}(\{\widetilde{\mathbf{N}}_X, \widetilde{\mathbf{A}}_X, \widetilde{\mathbf{w}}_X\}; \Theta_*), \quad (10)$$

where the GNN parameters $\Theta_*$ (Eq. (8)) are fixed. Comparing with Eq. (7), it is differentiable with regard to $\{\widetilde{\mathbf{N}}, \widetilde{\mathbf{A}}, \widetilde{\mathbf{w}}\}$ for all molecules in the neighborhood set $\mathcal{N}(X)$. Therefore, we can optimize the DST using gradient-based optimization method, e.g., an Adam optimizer (Kingma & Ba, 2015). The whole DST pipeline leverages iterative local discrete search (Equation 9, Algorithm 1), specifically, in $t$-th iteration, we optimize the DST of $X^{(t)}$, i.e., $X$ in Equation (10) is $X^{(t)}$.

**3.3.3 Sampling from DST** Then we sample the new scaffolding tree from the optimized DST. Concretely, (i) for each leaf node $v \in \mathcal{V}_{\text{leaf}}$ and the corresponding expansion node $u_v \in \mathcal{V}_{\text{expand}}$, we select one of the following step with probabilities (w.p.) as follows,

$$\mathcal{T} \sim \text{DST-Sampler}(\widetilde{\mathbf{N}}_*, \widetilde{\mathbf{A}}_*, \widetilde{\mathbf{w}}_*)$$

$$= \begin{cases} 1. \text{ SHRINK: delete leaf node } v, & \text{w.p. } 1 - (\widetilde{\mathbf{w}}_*)_v, \\ 2. \text{ EXPAND: add } u_v, \text{ select substructure at } u_v \text{ based on } (\widetilde{\mathbf{N}}_*)_{u_v}, & \text{w.p. } (\widetilde{\mathbf{w}}_*)_v (\widetilde{\mathbf{w}}_*)_{u_v|v}, \\ 3. \text{ REPLACE: select substructure at } v \text{ based on } (\widetilde{\mathbf{N}}_*)_u, & \text{w.p. } (\widetilde{\mathbf{w}}_*)_v (1 - (\widetilde{\mathbf{w}}_*)_{u_v|v}). \end{cases}$$
$$(11)$$

(ii) For each nonleaf node $v$, we expand a new node with probability $(\widetilde{\mathbf{w}}_*)_{u_v|v}$. If expanding, we select substructure at $u_v$ based on $(\widetilde{\mathbf{N}}_*)_{u_v}$. Otherwise, we "**do nothing**".

**3.3.4 Assemble** Each scaffolding tree corresponds to multiple molecules due to the multiple ways substructures can be combined. We enumerate all the possible molecules following Jin et al. (2018) (See Section C.5 in Appendix for details) for the further selection as described below.

### 3.4 MOLECULE DIVERSIFICATION

In the current iteration, we have generated $M$ molecules $(X_1, \cdots, X_M)$ and need to select $C$ molecules for the next iteration. We expect these molecules to have desirable chemical properties

(high $F$ score) and simultaneously maintain higher structural diversity. To do so, we resort to the *determinantal point process (DPP)* (Kulesza & Taskar, 2012), which models the repulsive correlation between data points. Specifically, for $M$ data points, whose indexes are $\{1, 2, \cdots, M\}$, $\mathbf{S} \in \mathbb{R}_+^{M \times M}$ denotes the similarity matrix between these data points. To create a diverse subset (denoted $\mathcal{R}$) with fixed size $C$, the sampling probability should be proportional to the determinant of the submatrix $\mathbf{S}_{\mathcal{R}} \in \mathbb{R}^{C \times C}$, i.e., $P(\mathcal{R}) \propto \det(\boldsymbol{S}_{\mathcal{R}})$, where $\mathcal{R} \subseteq \{1, 2, \cdots, M\}$, $|\mathcal{R}| = C$. Combining the objective ($F$) value and diversity, the composite objective is

$$\underset{\mathcal{R} \subseteq \{1,2,\cdots,M\}, |\mathcal{R}|=C}{\arg\max} \mathcal{L}_{\text{DPP}}(\mathcal{R}) = \lambda \sum_{r \in \mathcal{R}} F(X_r) + \log P(\mathcal{R}) = \log \det(\boldsymbol{V}_{\mathcal{R}}) + \log \det(\boldsymbol{S}_{\mathcal{R}}), \quad (12)$$

where the hyperparamter $\lambda > 0$ balances the two terms, the diagonal scoring matrix $\boldsymbol{V} = \text{diag}([\exp(\lambda F(X_1)), \cdots, \exp(\lambda F(X_M))])$, $\boldsymbol{V}_{\mathcal{R}} \in \mathbb{R}^{C \times C}$ is a sub-matrix of $\mathbf{V}$ indexed by $\mathcal{R}$. When $\lambda$ goes to infinity, it is equivalent to select top-$C$ candidates with the highest $F$ score, same as conventional evolutionary learning in (Jensen, 2019; Nigam et al., 2020). Inspired by generalized DPP methods (Chen et al., 2018), we further transform $\mathcal{L}_{\text{DPP}}(\mathcal{R})$, $\mathcal{L}_{\text{DPP}}(\mathcal{R}) = \log \det(\boldsymbol{V}_{\mathcal{R}}) + \log \det(\boldsymbol{S}_{\mathcal{R}}) = \log \det \left( \boldsymbol{V}_{\mathcal{R}}^{\frac{1}{2}} \boldsymbol{S}_{\mathcal{R}} \boldsymbol{V}_{\mathcal{R}}^{\frac{1}{2}} \right) = \log \det \left( (\boldsymbol{V}^{\frac{1}{2}} \mathbf{S} \boldsymbol{V}^{\frac{1}{2}})_{\mathcal{R}} \right)$, where $\boldsymbol{V}^{\frac{1}{2}} \mathbf{S} \boldsymbol{V}^{\frac{1}{2}}$ is symmetric positive semi-definite. Then it can be solved by generalized DPP methods in $O(C^2 M)$ (Chen et al., 2018) (Sec. F). The computational complexity of DST is $O(TMC^2)$ (Sec. C.8). Algorithm 1 summarizes the pipeline.

---

**Algorithm 1** Differentiable Scaffolding Tree (DST)

---

1: **Input**: Iteration $T$, beam width $C$, input molecule $X^{(1)}$. $\Phi = \{X^{(1)}\}$.
2: **Output**: Generated Molecule Set $\Omega$.
3: Learn GNN (Eq. 8): $\Theta_* = \arg\min_\Theta \sum_{(X,y) \in \mathcal{D}} \mathcal{L}(y, \widehat{y})$.    # Section 3.2.
4: **for** $t = 1, 2, \cdots, T$ **do**
5:    Initialize set $\Gamma = \{\}$.
6:    **for** $X^{(t)} \in \Phi$ **do**
7:       Initialize DST $\{\widetilde{\mathbf{N}}, \widetilde{\mathbf{A}}, \widetilde{\mathbf{w}}\}$ for $X^{(t)}$ (Eq. 2, 3, 4).
8:       Optimize DST: $\mathbf{N}_*, \mathbf{A}_*, \mathbf{w}_* = \arg\max_{\{\widetilde{\mathbf{N}}, \widetilde{\mathbf{A}}, \widetilde{\mathbf{w}}\}} \text{GNN}(\{\widetilde{\mathbf{N}}, \widetilde{\mathbf{A}}, \widetilde{\mathbf{w}}\}; \Theta_*)$ (Eq.10).
9:       Sample from DST: $\mathcal{T}_j^{(t+1)} \sim \text{DST-Sampler}(\mathbf{N}_*, \mathbf{A}_*, \mathbf{w}_*), j = 1, 2, \cdots$ (Eq.11).
10:       Assemble scaffolding tree $\mathcal{T}_j^{(t+1)}$ into molecules $X_j^{(t+1)}$. $\Gamma = \Gamma \cup \{X_j^{(t+1)}\}$.
11:    **end for**
12:    Select $\Phi \subseteq \Gamma, |\Phi| = C$ based on Eq. 12;   $\Omega = \Omega \cup \Phi$.    # Section 3.4
13: **end for**

---

# 4 EXPERIMENT

## 4.1 EXPERIMENTAL SETUP

**Molecular Properties** contains **QED**; **LogP**; **SA**; **JNK3**; **GSK3$\beta$**, following (Jin et al., 2020; Nigam et al., 2020; Moss et al., 2020; Xie et al., 2021), where QED quantifies drug-likeness; LogP indicates the water-octanol partition coefficient; SA stands for synthetic accessibility and is used to prevents the formation of chemically unfeasible molecules; JNK3/GSK3$\beta$ measure inhibition against c-Jun N-terminal kinase-3/Glycogen synthase kinase 3 beta. For all the 5 scores (including normalized SA), higher is better. We conducted (1) single-objective generation that optimizes JNK3, GSK3$\beta$ and LogP separately and (2) multi-objective generation that optimizes the mean value of "JNK3+GSK3$\beta$" and "QED+SA+JNK3+GSK3$\beta$" in the main text. Details are in Section C.3.

**Dataset**: ZINC 250K contains around 250K druglike molecules (Sterling & Irwin, 2015). We select the substructures that appear more than 1000 times in ZINC 250K as the vocabulary set $\mathcal{S}$, which contains 82 most frequent substructures. Details are in Section C.1. The code repository (including processed data, trained model, demonstration, molecules with the highest property) is available[2].

**Baselines**. (1) **LigGPT** (string-based distribution learning model with Transformer as a decoder) (Bagal et al., 2021); (2) **GCPN** (Graph Convolutional Policy Network) (You et al., 2018); (3) **MolDQN**

---

[2]https://github.com/futianfan/DST

(Molecule Deep Q-Network) (Zhou et al., 2019); (4) **GA+D** (Genetic Algorithm with Discriminator network) (Nigam et al., 2020); (5) **MARS** (Markov Molecular Sampling) (Xie et al., 2021); (6) **RationaleRL** (Jin et al., 2020); (7) **ChemBO** (Chemical Bayesian Optimization) (Korovina et al., 2020); (8) **BOSS** (Bayesian Optimization over String Space) (Moss et al., 2020); (9) **LSTM** (Long short term memory) (Brown et al., 2019); (10) **Graph-GA** (graph level genetic algorithm) (Brown et al., 2019). Among them, LigGPT belongs to the deep generative model, where all the oracle calls can be precomputed; GCPN, MolDQN are deep reinforcement learning methods; GA+D, MARS, Graph-GA are evolutionary learning methods; RationaleRL is a deep generative model fine-tuned with RL techniques. ChemBO and BOSS are Bayesian optimization methods. We also consider a DST variant: **DST-rand**. Instead of optimizing and sample from DST, DST-rand leverages random local search, i.e., randomly selecting basic operations (EXPAND, REPLACE, SHRINK) and substructure from the vocabulary. We also select a subset of all the random samples with high surrogate GNN prediction scores to improve efficiency. All the baselines except LigGPT require online oracle calls. Details are in Section B.

**Metrics**. For each method, we select top-100 molecules with highest property scores for evaluation, and consider the following metrics following Jin et al. (2018); You et al. (2018); Jin et al. (2020); Xie et al. (2021) (1) **Novelty (Nov)** (% of the generated molecules that are not in training set); (2) **Diversity (Div)** (average pairwise Tanimoto distance between the Morgan fingerprints); (3) **Average Property Score (APS)** (average score of top-100 molecules); (4) **# of oracle calls**: DST needs to call oracle in labeling data for GNN (**precomputed**) and DST based *de novo* generation (**online**), we show the costs for both steps. For each method in Table 1 and 2, we set the number of oracle calls so that the property score nearly converge w.r.t. oracle call's number. Details are in Section C.4.

Table 1: Multi-objective *de novo* design. #oracle = (1)"**precomputed** oracle call" (to label molecules in existing database) + (2)"**online** oracle call" (during learning).

| Method | JNK3+GSK3$\beta$ | | | | QED+SA+JNK3+GSK3$\beta$ | | | |
|---|---|---|---|---|---|---|---|---|
| | Nov↑ | Div↑ | APS↑ | #oracle↓ | Nov↑ | Div↑ | APS↑ | #oracle↓ |
| LigGPT | **100%** | **0.845** | 0.271 | 100k+0 | **100%** | **0.902** | 0.378 | 100k+0 |
| GCPN | **100%** | 0.578 | 0.293 | 0+200K | **100%** | 0.596 | 0.450 | 0+200K |
| MolDQN | **100%** | 0.605 | 0.348 | 0+200K | **100%** | 0.597 | 0.365 | 0+200K |
| GA+D | **100%** | 0.657 | 0.608 | 0+50K | 97% | 0.681 | 0.632 | 0+50K |
| RationaleRL | **100%** | 0.700 | 0.795 | 25K+67K | 99% | 0.720 | 0.675 | 25K+67K |
| MARS | **100%** | 0.711 | 0.789 | 0+50K | **100%** | 0.714 | 0.662 | 0+50K |
| ChemBO | 98% | 0.702 | 0.747 | 0+50K | 99% | 0.701 | 0.648 | 0+50K |
| BOSS | 99% | 0.564 | 0.504 | 0+50K | 98% | 0.561 | 0.504 | 0+50K |
| LSTM | **100%** | 0.712 | 0.680 | 0+30K | **100%** | 0.706 | 0.672 | 0+30K |
| Graph-GA | **100%** | 0.634 | 0.825 | 5+25K | **100%** | 0.723 | 0.714 | 5+25K |
| DST-rand | **100%** | 0.456 | 0.622 | 10+5K | **100%** | 0.765 | 0.575 | 20K+5K |
| DST | **100%** | 0.750 | **0.827** | **10K+5K** | 100% | 0.755 | **0.752** | **20K+5K** |

## 4.2 OPTIMIZATION PERFORMANCE

The results of multi-objective and single-objective generation are shown in Table 1 and 2. We find that DGM (LigGPT) and RL-based methods (GCPN and MolDQN) fail in some tasks, which is consistent with the results reported in RationaleRL (Jin et al., 2020) and MARS (Xie et al., 2021). Overall, DST obtains the best results in most tasks. In terms of success rate and diversity, DST outperformed all baselines in most tasks. It also reached the highest scores within $T = 50$ iterations in most optimization tasks (see Table 5 and 6 in Appendix). Especially in optimizing LogP, the model successfully learned to add a six-member ring (see Figure 9 in Appendix) each step, which is theoretically the optimal strategy under our setting. Combined with the ablation study compared with random selection (see Figure 12 in Appendix), our results show the local gradient defined by DST is a useful direction indicator, consistent with the concept of gradient. Further, achieving high diversity validates the effect of the DPP-based selection strategy. Although the novelty is not the highest, it is still comparable to baseline methods. These results show our gradient-based optimization strategy has a strong optimization ability to provide a diverse set of molecules with high objective functions.

## 4.3 ORACLE EFFICIENCY

As mentioned above, oracle calls for realistic optimization tasks can be time-consuming and expensive. From Table 1 and 2, we can see that majority of de novo optimization methods require oracle calls online (instead of precomputation), including all of RL/evolutionary algorithm based baselines. DST takes fewer oracle calls compared with baselines. DST can leverage the precomputed oracle calls to label the molecules in an existing database (i.e., ZINC) for training the oracle GNN and dramatically saving the oracle calls during reference. In the three tasks in Table 2, two-thirds of the oracle calls

Table 2: Single-objective *de novo* molecular generation.

| Method | JNK3 | | | | GSK3$\beta$ | | | | LogP | | | |
|---|---|---|---|---|---|---|---|---|---|---|---|---|
| | Nov↑ | Div↑ | APS↑ | #oracle↓ | Nov↑ | Div↑ | APS↑ | #oracle↓ | Nov↑ | Div↑ | APS↑ | #oracle↓ |
| LigGPT | **100%** | **0.837** | 0.302 | 100K+0 | **100%** | **0.867** | 0.283 | 100K+0 | **100%** | **0.868** | 4.56 | 100K+0 |
| GCPN | **100%** | 0.584 | 0.365 | 0+200K | **100%** | 0.519 | 0.400 | 0+200K | **100%** | 0.532 | 5.43 | 0+200K |
| MolDQN | **100%** | 0.605 | 0.459 | 0+200K | **100%** | 0.545 | 0.398 | 0+200K | **100%** | 0.485 | 6.00 | 0+200K |
| GA+D | 99% | 0.702 | 0.615 | 0+50K | 98% | 0.687 | 0.678 | 0+50K | **100%** | 0.721 | 30.2 | 0+50K |
| RationaleRL | 99% | 0.681 | 0.803 | 25K+32K | 99% | 0.731 | 0.806 | 30K+45K | - | - | - | - |
| MARS | **100%** | 0.711 | 0.784 | 0+50K | **100%** | 0.735 | 0.810 | 0+50K | **100%** | 0.692 | 44.1 | 0+30K |
| ChemBO | 98% | 0.645 | 0.648 | 0+50K | 98% | 0.679 | 0.492 | 0+50K | 98% | 0.732 | 10.2 | 0+50K |
| BOSS | 98% | 0.601 | 0.471 | 0+50K | 99% | 0.658 | 0.432 | 0+50K | 100% | 0.735 | 9.64 | 0+50K |
| LSTM | **100%** | 0.745 | 0.645 | 0+20K | **100%** | 0.715 | 0.674 | 0+20K | **100%** | 0.702 | 10.1 | 0+30K |
| Graph-GA | **100%** | 0.674 | 0.868 | 5+10K | **100%** | 0.705 | **0.875** | 5+10K | **100%** | 0.721 | 45.0 | 5+10K |
| DST-rand | **100%** | 0.754 | 0.413 | 10K+10K | 97% | 0.793 | 0.455 | 10K+10K | **100%** | 0.713 | 36.1 | 10K+15K |
| DST | **100%** | 0.732 | **0.928** | **10K+5K** | **100%** | 0.748 | 0.869 | **10K+5K** | **100%** | 0.704 | **47.1** | **10K+5K** |

(10K) can be precomputed or collected from other sources. To further verify the oracle efficiency, we explore a special setting of molecule optimization where the budget of oracle calls is limited to a fixed number (2K, 5K, 10K, 20K, 50K) and compare the optimization performance. For GCPN, MolDQN, GA+D, and MARS, the learning iteration number depends on the budget of oracle calls. RationaleRL (Jin et al., 2020) is omitted because it requires intensive oracle calls to collect enough reference data, exceeding the oracle budget in this scenario. In DST, we use around 80% budget to label the dataset (i.e., training GNN) while the remaining budget to conduct *de novo* design. Specifically, for 2K, 5K, 10K, 20K, 50K, we use 1.5K, 4K, 8K, 16K, and 40K oracle calls to label the data for learning GNN, respectively. We show the average objective values of top-100 molecules under different oracle budgets in Figure 3. Our method has a significant advantage compared to all the baseline methods in limited budget settings. We conclude that supervised learning is a well-studied and much easier task than generative modeling.

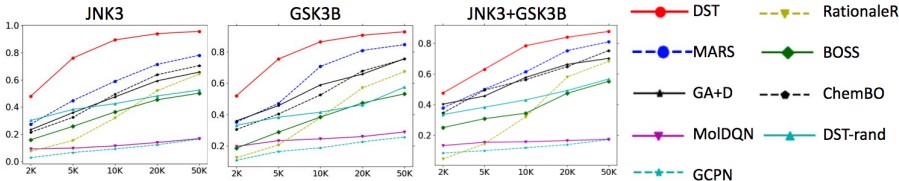

Figure 3: Oracle efficiency test. Top-100 average score v.s. the number of oracle calls.

## 4.4 INTERPRETABILITY ANALYSIS

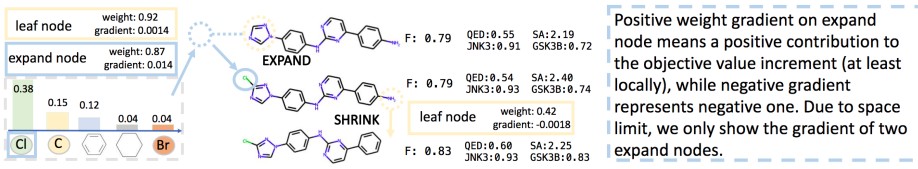

Figure 4: Two steps in optimizing "QED+SA+JNK3+GSK3$\beta$".

To obtain more insights from the local gradient, We visualize two modification steps in Figure 4. The node weights and gradient values interpret the property at the substructure level. This is similar to most attribution-based interpretable ML methods, e.g., saliency map (Simonyan et al., 2013).

## 5 CONCLUSION

This paper proposed differentiable scaffolding tree (DST) to make a molecular graph locally differentiable, allowing a continuous gradient-based optimization. To the best of our knowledge, it is the first attempt to make the molecular optimization problem differentiable at the substructure level, rather than resorting to latent spaces or using RL/evolutionary algorithms. We constructed a general molecular optimization strategy based on DST, corroborating thorough empirical studies.

## REPRODUCIBILITY STATEMENT

The code repository is available at `https://github.com/futianfan/DST`, including README file, all the codes for data preprocessing, model architecture, molecules with the highest property. All the data we use are publicly available. Section B describes an experimental setup for all the baseline methods. We elaborate on the implementation details in Section C. Concretely, Section C.1 describes the dataset we use; Section C.2 describes software and hardware configuration; Section C.3 introduces the target molecular properties to optimize; Section C.4 describes all the metrics for evaluating the performance; Section C.6 describes the setup for model implementation, especially hyperparameter setup.

## ACKNOWLEDGEMENT

This work was supported by NSF award SCH-2014438, PPoSS 2028839, IIS-1838042, NIH award R01 1R01NS107291-01, OSF Healthcare, ONR under grant number N00014-21-1-2195, and the MIT UROP office.

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

CONTENTS

APPENDIX TO DIFFERENTIABLE SCAFFOLDING TREE FOR MOLECULAR OPTIMIZATION

The appendix is organized as follows. First, we list the complete mathematical notations for ease of exposition. Then, we show additional experimental setup and empirical results, including baseline setup in Section B, implementation details of our method in Section C, additional experimental results in Section D. Then, we provide theoretical analysis in Section E, extend molecule diversification in Section, and prove the theoretical results in the main paper in Section G.

Table 3: Complete Mathematical Notations.

| Notations | Descriptions |
|---|---|
| $\mathcal{O}$ | Oracle function, e.g., evaluator of molecular property (Def 1). |
| $F$ | objective function of molecule generation (Eq. 1). |
| $P \in \mathbb{N}_+$ | Number of target oracles. |
| $\mathcal{Q}$ | Set of all the valid chemical molecules. |
| $\mathcal{S}$ | Vocabulary set, i.e., substructure set. A substructure is an atom or a ring. |
| $\mathcal{T}$ | Scaffolding tree (Def 3). |
| $K = |\mathcal{T}|$ | number of nodes in scaffolding tree $\mathcal{T}$. |
| $\mathbf{N}; \mathbf{A}; \mathbf{w}$ | Node indicator matrix; adjacency matrix; node weight. |
| $\mathcal{V}_{\text{leaf}}$ | Leaf node set in scaffolding tree $\mathcal{T}$. |
| $\mathcal{V}_{\text{nonleaf}}$ | Nonleaf node set in scaffolding tree $\mathcal{T}$. |
| $\mathcal{V}_{\text{expand}}$ | Expansion node set in scaffolding tree $\mathcal{T}$. |
| $K_{\text{leaf}} = |\mathcal{V}_{\text{leaf}}|$ | Size of leaf node set. |
| $K_{\text{expand}} = |\mathcal{V}_{\text{expand}}| = K$ | Size of expansion node set. $K_{\text{leaf}} = K_{\text{expand}}$. |
| $d \in \mathbb{N}_+$ | GNN hidden dimension. |
| $L \in \mathbb{N}_+$ | GNN depth. |
| $\Theta = \{\mathbf{E}\} \cup \{\mathbf{B}^{(l)}, \mathbf{U}^{(l)}\}_{l=1}^{L}$ | Learnable parameter of GNN. |
| $\mathbf{E} \in \mathbb{R}^{|\mathcal{S}| \times d}$ | embedding stackings of all the substructures in vocabulary set $\mathcal{S}$. |
| $\mathbf{B}^{(l)} \in \mathbb{R}^{K \times d}$ | bias parameters at $l$-th layer. |
| $\mathbf{U}^{(l)} \in \mathbb{R}^{d \times d}$ | weight parameters at $l$-th layer. |
| $\mathbf{H}^{(l)}, l = 0, \cdots, L$ | Node embedding at $l$-th layer of GNN |
| $\mathbf{H}^{(0)} = \mathbf{NE} \in \mathbb{R}^{K \times d}$ | initial node embeddings, stacks basic embeddings of all the nodes in the scaffolding tree. |
| MLP | multilayer perceptron |
| ReLU | ReLU activate function |
| $\widehat{y}$ | GNN prediction. |
| $y$ | groundtruth |
| $\mathcal{L}$ | Loss function of GNN. |
| $\mathcal{D}$ | the training set |
| $\mathcal{N}(X)$ | Neighborhood molecule set of $X$ (Def 10). |
| $\Lambda$ | differentiable edge set. |
| $\widetilde{\mathbf{N}}; \widetilde{\mathbf{A}}; \widetilde{\mathbf{w}}$ | Differentiable node indicator matrix; adjacency matrix; node weight. |
| $\det()$ | Determinant of a square matrix |
| $M \in \mathbb{N}_+$ | Number of all possible molecules to select. |
| $C \in \mathbb{N}_+$ | Number of selected molecules. |
| $\mathbf{S} \in \mathbb{R}_+^{M \times M}$ | Similarity kernel matrix. |
| $\mathbf{V} \in \mathbb{R}_+^{M \times M}$ | Diagonal scoring matrix. |
| $\mathcal{R}$ | subset of $\{1, 2, \cdots, M\}$, index of select molecules. |
| $\lambda > 0$ | hyperparameter in Eq. 12 and 17, balances desirable property and diversity. |

## A    COMPLETE MATHEMATICAL NOTATIONS.

In this Section, we show all the mathematical notations in Table 3 for completeness.

We also illustrate the difference between DST and existing methods in Figure 5.

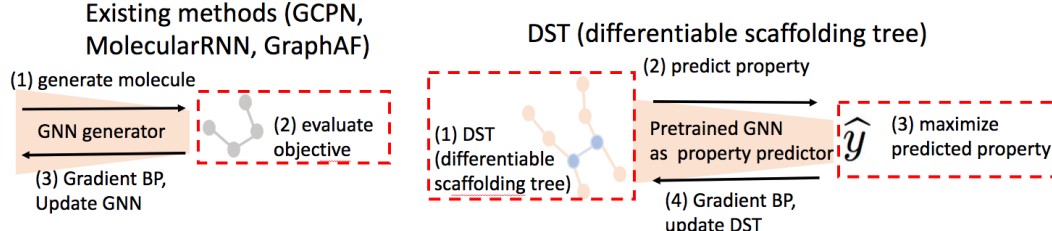

Figure 5: **Left**: Most of the existing methods (including GCPN (You et al., 2018), Molecular-RNN (Popova et al., 2019), GraphAF (Shi et al., 2020)) use GNN as a graph generator. Specifically, (1) generate molecule; (2) evaluate learning objective (loss in deep generative model or reward in reinforcement learning); (3) back-propagate (BP) gradient to **update GNN**. In sum, the learning objective is differentiable w.r.t. the GNN's parameters.
**Right**: Regarding DST, given the pretrained GNN as surrogate oracle model (i.e., property predictor), we have several steps: (1) construct differentiable scaffolding tree (DST); (2) predict the property via GNN; (3) maximize the predicted property $\hat{y}$ (learning objective); (4) back-propagate (BP) gradient to **update DST**. In sum, the learning objective is differentiable w.r.t. DST (also input of GNN).
**Summary**: DST makes the learning objective differentiable w.r.t molecule graph structure, while prior works make the learning objective differentiable w.r.t. neural networks' parameters. Our approach directly optimizes molecular graph structures, while prior works indirectly search for the molecule graphs with the help of a neural network.

## B    BASELINE SETUP

In this section, we describe the experimental setting for baseline methods. Most of the settings follow the original papers.

- **LigGPT** (string-based distribution learning model with Transformer as a decoder) (Bagal et al., 2021) is trained for 10 epochs using the Adam optimizer with a learning rate of $6e - 4$. LigGPT comprises stacked decoder blocks, each of which is composed of a masked self-attention layer and fully connected neural network. Each self-attention layer returns a vector of size 256 that is taken as input by the fully connected network. The hidden layer of the neural network outputs a vector of size 1024 and uses a GELU activation, and the final layer again returns a vector of size 256 to be 7 used as input for the next decoder block. LigGPT consists of 8 such decoder blocks. LigGPT has around 6M parameters.

- **GCPN** (Graph Convolutional Policy Network) (You et al., 2018) leveraged graph convolutional network and policy gradient to optimize the reward function that incorporates target molecular properties and adversarial loss. In each step, the allowable action to the current molecule could be either connecting a new substructure or an atom with an existing molecular graph or adding a bond to connect existing atoms. GCPN predicts the actions and is trained via proximal policy optimization (PPO) to optimize an accumulative reward, including molecular property objectives and adversarial loss. Both policy network and adversarial network (discriminative training) use the same neural architecture, which is a three-layer graph convolutional network (GCN) (Kipf & Welling, 2016) with 64 hidden nodes. Batch normalization is adopted after each layer, and sum-pooling is used as the aggregation function. Adam optimizer is used with 1e-3 initial learning rate, and batch size is 32.

- **MolDQN** (Molecule Deep Q-Networks) (Zhou et al., 2019), same as GCPN, formulate the molecule generation procedure as a Markov Decision Process (MDP) and use Deep Q-Network to solve it. The reward includes target property and similarity constraints.

Following the original paper, the episode number is 5,000, maximal step in each episode is 40. Each step calls oracle once; thus, 200K oracle calls are needed in one generation process. The discount factor is 0.9. Deep Q-network is a multilayer perceptron (MLP) whose hidden dimensions are 1024, 512, 128, 32, respectively. The input of the Q-network is the concatenation of the molecule feature (2048-bit Morgan fingerprint, with a radius of 3) and the number of left steps. Adam is used as an optimizer with 1e-4 as the initial learning rate. Only rings with a size of 5 and 6 are allowed. It leverages $\epsilon$-greedy together with randomized value functions (bootstrapped-DQN) as an exploration policy, $\epsilon$ is annealed from 1 to 0.01 in a piecewise linear way.

- **GA+D** (Genetic Algorithm with Discriminator network) (Nigam et al., 2020) uses a deep neural network as a discriminator to enhance exploration in a genetic algorithm. $\beta$ is an important hyperparameter that weights the importance of the discriminator's loss in the overall fitness function, and we set it to 10. The generator runs 100 generations with a population size of 100 for *de novo* molecular optimization and 50 generations with a population size of 50 for molecular modification. Following the original paper (Nigam et al., 2020), the architecture of the discriminator is a two-layer fully connected neural network with ReLU activation and a sigmoid output layer. The hidden size is 100, while the size of the output layer is 1. The input feature is a vector of chemical and geometrical properties characterizing the molecules. We used Adam optimizer with 1e-3 as the initial learning rate.

- **RationaleRL** (Jin et al., 2020) is a deep generative model that grows a molecule atom-by-atom from an initial rationale (subgraph). The architecture of the generator is a message-passing network (MPN) followed by MLPs applied in breadth-first order. The generator is pre-trained on general molecules combined with an encoder and then fine-tuned to maximize the policy gradient reward function. The encoder and decoder MPNs both have hidden dimensions of 400. The dimension of the latent variable is 20. Adam optimizer is used on both pre-training and fine-tuning with initial learning rates of 1e-3, 5e-4, respectively. The annealing rate is 0.9. We pre-trained the model with 20 epochs.

- **MARS** (Xie et al., 2021) leverage Markov chain Monte Carlo sampling (MCMC) on molecules with an annealing scheme and an adaptive proposal. The proposal is parameterized by a graph neural network trained on MCMC samples. We follow most of the settings in the original paper. The message passing network has six layers, where the node embedding size is set to 64. Adam is used as an optimizer with 3e-4 initial learning rate. To generate a basic unit, top-1000 frequent fragments are drawn from ZINC database (Sterling & Irwin, 2015) by enumerating single bonds to break. During the annealing process, the temperature $T = 0.95^{\lfloor t/5 \rfloor}$ would gradually decrease to 0.

- **ChemBO** (chemical Bayesian optimization) (Korovina et al., 2020) leverage Bayesian optimization. It also explores the synthesis graph in a sample-efficient way and produces synthesizable candidates. Following the default setting in the original paper, the number of steps of acquisition optimization is set to 20. The initial pool size is set to 20, while the maximal pool size is 1000. Regarding the kernel, we leveraged the kernel proposed in ChemBO, which is the optimal-transport-based distance and kernel that accounts for graphical information explicitly. We did extra experiments to validate that it empirically outperforms the Tanimoto similarity kernel based on the molecular fingerprint. As for the acquisition, we followed the original paper and adopted the ensemble method using the EI (Expected Improvement), UCB (Upper Confidence Bound), and TTEI (Top-Two Expected Improvement) acquisitions; we conducted an extra experiment and found it empirically outperforms a single acquisition in our scenario, which is consistent to the ChemBO paper. GPs do not scale well with big data. To address this issue, we use Subset of DataChalupka et al. (2013), which uses a subset of all the data points to reduce the size of the kernel matrix and learn the surrogate model efficiently. Following Chalupka et al. (2013), we selected a subset of data points using k-means clustering based on molecular fingerprint and sampled data points from each cluster. The size of the subset is set to $m = 1000$. The computational complexity is $O(m^3)$.

- **BOSS** (Bayesian Optimization over String Space) (Moss et al., 2020) builds a Gaussian process surrogate model based on Sub-sequence String Kernel (SSK)[3], which naturally supports SMILES strings with variable length, and maximizing acquisition function efficiently

---

[3] based on SMILES string

for spaces with syntactical constraints. Following their default setting, we only keep the SMILES string whose length is less than 50 (except the LogP task). At each BO step, BOSS samples 100 candidates, querying those that maximize the acquisition function predicted by SSK. It utilized a genetic algorithm (GA) to optimize acquisition function efficiently under syntactical constraints. We empirically compared GA based acquisition optimizer with random search based acquisition optimizer (provided in the BOSS code repository) to validate its superiority. GA requires intensive oracle calls and is leveraged in inner-loop maximization in BOSS, so we do not spend too much computational resources in a single GA step. The population size is set to 100, the generation (evolution) number is set to 100. These setups are nearly optimal, and tuning these hyperparameters didn't improve the performance.

- **DST-rand** a `DST` variant, which randomly selects basic operations (EXPAND, REPLACE, SHRINK) and substructure from vocabulary). During each iteration, it randomly selects basic operations (EXPAND, SHRINK, REPLACE) and substructure from vocabulary to generate at most $K_1$ molecules, select at most top-$K_2$ molecules with highest GNN surrogate predictions. Then we evaluate the $K_2$ candidates with the real oracle and select $C$ molecules as starting points for the next iteration. For a fair comparison, $C$ is set to 10, same as `DST`. To get the near-optimal setup, we try different combinations of $(K_1, K_2)$, and finally find that $K_1 = 1000, K_2 = 100$ achieves the best performance in terms of the average of top-100 molecules' score. Other setups of DST-rand follow `DST`.

## C   IMPLEMENTATION DETAILS

### C.1   DATASET

We use ZINC 250K dataset, which contains around 250K druglike molecules extracted from the ZINC database (Sterling & Irwin, 2015). We first cleaned the data by removing the molecules containing out-of-vocabulary substructure and having 195K molecules left.

**Vocabulary $\mathcal{S}$: set of substructure**. The substructure is the basic building block in our method, including frequent atoms and rings. On the other hand, atom-wise molecule generation is tricky due to the existence of rings. To select the substructure set $\mathcal{S}$, we break all the ZINC molecules into substructures (including single rings and single atoms), count their frequencies, and include the substructures whose frequencies are higher than 1000 into vocabulary set $\mathcal{S}$.

The final vocabulary contains 82 substructures, including the frequent atoms like carbon atom, an oxygen atom, nitrogen atom, and frequent rings like benzene ring. The vocabulary set covers over 80% of molecules in ZINC databases. After removing the molecules that contain out-of-vocabulary substructure, we use a random subset of the remaining molecules to train the GNNs, depending on the oracle budget.

The vocabulary size is big enough for this proof-of-concept study. Other works also need to constrain their design space, such as MolDQN only allowing three types of atoms in a generation: "C", "N", "O" (Zhou et al., 2019); JTVAE (Jin et al., 2018), and RationaleRL (Jin et al., 2020) only using frequent substructures similar to our setting. On the other hand, we may not want infrequent atoms or substructures because rare substructures in ZINC may have some undesired properties such as toxicity, may not be stable, may not be easily synthesizable (Gao & Coley, 2020). Also, rare substructures may impede the learning of oracle GNN. Note that users can enlarge the substructure space when they apply our method. We show all the 82 substructures in $\mathcal{S}$ in Figure 6.

### C.2   SOFTWARE/HARDWARE CONFIGURATION

We implemented `DST` using Pytorch 1.7.0, Python 3.7, RDKit v2020.09.1.0 on an Intel Xeon E5-2690 machine with 256G RAM and 8 NVIDIA Pascal Titan X GPUs.

### C.3   TARGET MOLECULAR PROPERTIES

Target molecular properties include

- **QED** represents a quantitative estimate of drug-likeness. QED score ranges from 0 to 1. It can be evaluated by the RDKit package (https://www.rdkit.org/).

- **LogP** represents octanol-water partition coefficient, measuring molecules' solubility. LogP score ranges from $-\infty$ to $+\infty$. Thus, when optimizing LogP individually, we use the GNN model to do regression.

- **SA** (Synthetic Accessibility) score measures how hard it is to synthesize a given molecule, based on a combination of the molecule's fragments contributions (Ertl & Schuffenhauer, 2009). It is evaluated via RDKit (Landrum et al., 2006). The raw SA score ranges from 1 to 10. A higher SA score means the molecule is hard to be synthesized and is not desirable. In the multiple-objective optimization, we normalize the SA score to $[0, 1]$ so that a higher normalized SA value is easy to synthesize. Following (Gao & Coley, 2020), we use the normalize function for raw SA score,

$$\text{normalized-SA}(X) = \begin{cases} 1, & \text{SA}(X) < \mu \\ \exp\left(-\frac{(\text{SA}(X)-\mu)^2}{2\sigma^2}\right), & \text{SA}(X) \geq \mu, \end{cases}$$

  where $\mu = 2.230044, \sigma = 0.6526308$.

- **JNK3** (c-Jun N-terminal Kinases-3) belongs to the mitogen-activated protein kinase family and are responsive to stress stimuli, such as cytokines, ultraviolet irradiation, heat shock, and osmotic shock. Similar to GSK3$\beta$, JNK3 is also evaluated by well-trained[4] random forest classifiers using ECFP6 fingerprints using ExCAPE-DB dataset (Li et al., 2018; Jin et al., 2020), and the range is also $[0, 1]$.

- **GSK3$\beta$** (Glycogen synthase kinase 3 beta) is an enzyme that in humans is encoded by the GSK3$\beta$ gene. Abnormal regulation and expression of GSK3$\beta$ is associated with an increased susceptibility towards bipolar disorder. It is evaluated by well-trained[5] random forest classifiers using ECFP6 fingerprints using ExCAPE-DB dataset (Li et al., 2018; Jin et al., 2020). GSK3$\beta$ score of a molecule ranges from 0 to 1.

For QED, LogP, normalized SA, JNK3, and GSK3$\beta$, higher scores are more desirable under our experimental setting.

## C.4 EVALUATION METRICS

We leverage the following evaluation metrics to measure the optimization performance:

- **Novelty** is the fraction of the generated molecules that do not appear in the training set.

- **Diversity** of generated molecules is defined as the average pairwise Tanimoto distance between the Morgan fingerprints (You et al., 2018; Jin et al., 2020; Xie et al., 2021).

$$\text{diversity} = 1 - \frac{1}{|\mathcal{Z}|(|\mathcal{Z}| - 1)} \sum_{Z_1, Z_2 \in \mathcal{Z}, Z_1 \neq Z_2} \text{sim}(Z_1, Z_2), \tag{13}$$

  where $\mathcal{Z}$ is the set of generated molecules. $\text{sim}(Z_1, Z_2)$ is the Tanimoto similarity between molecule $Z_1$ and $Z_2$.

- **(Tanimoto) Similarity** measures the similarity between the input molecule and generated molecules. It is defined as

$$\text{sim}(X, Y) = \frac{\mathbf{b}_X^\top \mathbf{b}_Y}{\|\mathbf{b}_X\|_2 \|\mathbf{b}_Y\|_2},$$

  $\mathbf{b}_X$ is the binary Morgan fingerprint vector for the molecule $X$. In this paper, it is a 2048-bit binary vector.

- **SR** (Success Rate) is the percentage of the generated molecules that satisfy the property constraint measured by objective $f$ defined in Equation (1). For single-objective *de novo* molecular generation, the objective $f$ is the property score, the constraints for JNK3,

---

[4]The test AUROC score is 0.86 (Jin et al., 2020).
[5]The test AUROC score is also 0.86 (Jin et al., 2020).

GSK3$\beta$ and LogP are JNK3$\geq 0.5$, GSK3$\beta \geq 0.5$ and LogP$\geq 5.0$ respectively. For multi-objective *de novo* molecular generation, the objective $f$ is the average of all the normalized target property scores. Concretely, when optimizing "JNK3+GSK3$\beta$", both JNK3 and GSK3$\beta$ ranges from 0 to 1, $f$ is average of JNK3 and GSK3$\beta$ scores; when optimizing "QED+SA+JNK3+GSK3$\beta$", we first normalized SA to 0 to 1. $f$ is average of QED, normalized SA, JNK3 and GSK3$\beta$ scores. The constraint is the $f$ score is greater than 0.4.

- **# of oracle calls** during the generation process. `DST` needs to call oracle in labeling data for GNN and `DST` based *de novo* generation, thus we show the costs for both steps.

- **chemical validities**. As we only enumerate valid chemical structures during the recovery from scaffolding trees (Section C.5), the chemical validities of the molecules produced by `DST` are always 100%.

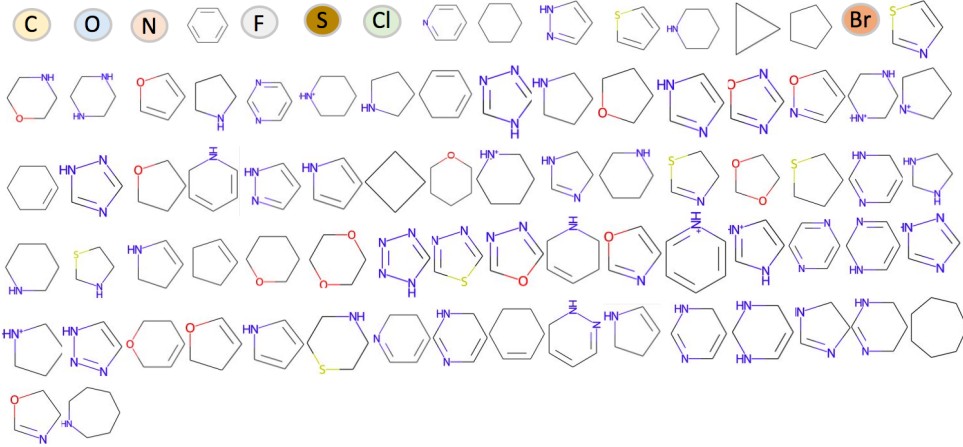

Figure 6: All the substructures in the vocabulary set $\mathcal{S}$, drawn from ZINC 250K database (Sterling & Irwin, 2015). It includes atoms and single rings appearing more than 1000 times in the ZINC250K database.

## C.5 Assembling Molecule from Scaffolding Tree

Each scaffolding tree corresponds to multiple molecules due to rings' multiple combination ways. For each scaffolding tree, we enumerate all the possible molecules following Jin et al. (2018) for further selection. We provide two examples in Figure 7 to illustrate it. Two examples are related to ring-atom combination and ring-ring combination, respectively. For ring-ring combination, our current setting does not support the spiro compounds (contains rings sharing one atom but no bonds) or phenalene-like compounds (contains three rings sharing one atom, and each two of them sharing a bond). These two cases are relatively rare chemical structures in the context of drug discovery (Supsana et al., 2005). As we only enumerate valid chemical structures during the recovery from scaffolding trees, the chemical validities are always 100%.

## C.6 Details on GNN Learning and DST Optimization

Both the size of substructure embedding and hidden size of GCN (GNN) in Eq. (6) are $d = 100$. The depth of GNN $L$ is 3. When training GNN, the training epoch number is 5, and we evaluate the loss function on the validation set every 20K data passes. When the validation loss would not decrease, we terminate the training process. When optimizing "JNK3", "GSK3$\beta$", "QED", "JNK3+GSK3$\beta$" and "QED+SA+JNK3+GSK3$\beta$", we use binary cross entropy as loss criterion. When optimizing "LogP", since LogP ranges from $-\infty$ to $+\infty$, we leverage GNN to conduct regression tasks, and use means square error (MSE) as loss criteria $\mathcal{L}$. In *the de novo* generation, in each generation, we keep $C = 10$ molecules for the next iteration. In most cases in the experiment, the size of the neighborhood set (Definition. 10) is less than 100. We use Adam optimizer with 1e-3 learning rate in training and inference procedure, optimizing the GNN and differentiable scaffolding tree, respectively. When optimizing `DST`(Equation 10), our method processes one `DST` at a time, we conduct 1000 Adam steps

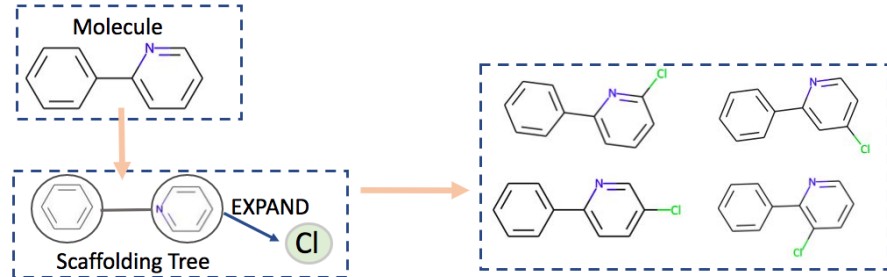

(a) Ring-atom connection. When connecting atom and ring in a molecule, an atom can be connected to any possible atoms in the ring. In the example, there are 4 possible ways to add a Chlorine atom ("Cl") as an expansion node to the target ring, a leaf node in the scaffolding tree.

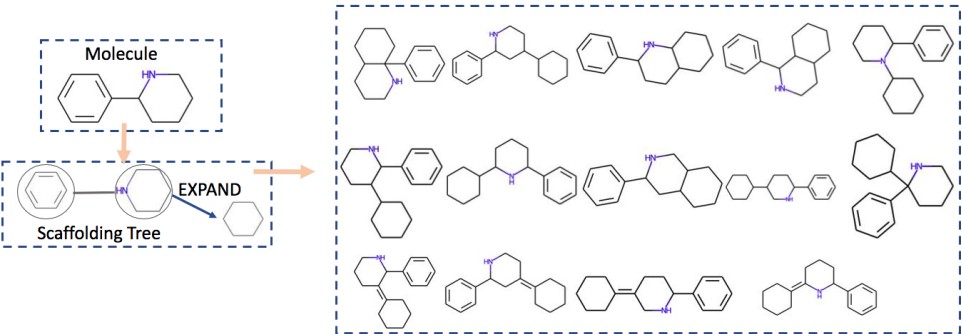

(b) Ring-ring connection. When connecting ring and ring, there are two general ways, (1) one is to use a bond (single, double, or triple) to connect the atoms in the two rings. (2) another is two rings share two atoms and one bond. In the example, there are 14 possible ways to add a Cyclohexane ring (SMILES is "C1CCCCC1") and connect it to the target ring, a leaf node in the scaffolding tree.

Figure 7: Assemble examples.

in every iteration, which is sufficient to converge in almost all the cases. As a complete generation algorithm, we optimize a batch parallelly and select candidates based on DPP. When we use up oracle budgets, we stop it. All DST results in the tables take at most $T = 50$ iterations.

## C.7 RESULTS OF DIFFERENT RANDOM SEEDS

This section presents the empirical results that use different random seeds for multiple runs. In our pipeline, the random error comes from in two steps: (1) Training oracle GNN: data selection/split, training process including data shuffle and GNN's parameter initialization. (2) Inference (Optimizing DST): before optimizing DST, we initialize the learnable parameter randomly, including $\widetilde{\mathbf{N}}$, $\widetilde{\mathbf{w}}$, $\widetilde{\mathbf{A}}$, which also brings randomness. To measure the robustness of the proposed method, we use 5 different random seeds for the whole pipeline and compare the difference of 5 independent trials. The results are reported in Table 4. We find that almost all the metrics would not change significantly among various trials, validating the robustness of the proposed method.

Table 4: Results of 5 independent trials using different random seeds. For novelty, diversity, and SR (success rate), we report the average value of 5 runs and their standard deviation.

| Tasks | Novelty↑ | Diversity↑ | SR↑ | # Oracles↓ |
|---|---|---|---|---|
| JNK3 | 98.1%±0.3% | 0.722±0.032 | 92.8%±0.5% | 10K+5K |
| GSK3$\beta$ | 98.6%±0.5% | 0.738±0.047 | 91.8%±0.3% | 10K+5K |
| LogP | 100.0%±0.0% | 0.716±0.032 | 100.0%±0.0% | 10K+5K |
| JNK3+GSK3$\beta$ | 98.6%±1.1% | 0.721±0.021 | 91.3%±0.6% | 10K+5K |
| QED+SA+JNK3+GSK3$\beta$ | 99.2%±0.3% | 0.731±0.029 | 79.4%±1.2% | 20K+5K |

### C.8 COMPLEXITY ANALYSIS

We did a computational analysis in terms of oracle calls and computational complexity. (1) **oracle calls**. DST requires $O(TM)$ oracle calls, where $T$ is the number of iterations (Alg 1). $M$ is the number of generated molecules (Equation. 12), we have $M \leq KJ$, $K$ is the number of nodes in the scaffolding tree, for small molecule, $K$ is very small. $J$ is the number of enumerated candidates in each node. As shown in Figure 7, $J$ is also upper-bounded ($J \leq 4 + 14$ for the example in Figure 7). (2) **computational complexity**. The computational complexity is $O(TMC^2)$ (the main bottleneck is DPP method, Algorithm 2), where the size of selected molecules $C = 10$ for all the tasks (Section 3.4 & C.6). For all the tasks in Table 2 and 1, DST can be finished in 12 hours on an Intel Xeon E5-2690 562 machine with 256G RAM and 8 NVIDIA Pascal Titan X GPUs. The complexity and runtime are acceptable for molecule optimization.

## D ADDITIONAL EXPERIMENTAL RESULTS

This section presents the additional empirical results, including additional results on *de novo* generation, ablation study, chemical space visualization, and interpretability analysis (case study).

### D.1 ADDITIONAL RESULTS OF *de novo* MOLECULAR GENERATION

This section presents some additional results of *de novo* molecular generation for completeness.

First, we present the optimization curve for all the optimization tasks in Figure 8. We observe that our method can efficiently reach a high objective value within 10 iterations in all the optimization tasks. Worth mentioning that when optimizing LogP, the model successfully learned to add a six-member ring each step, as shown in Figure 9, and the objective ($F$) value grows linearly as a function of iteration number, which is theoretically the optimal strategy under our setting. Then, in Figure 10, we show the molecules with the highest objective ($F$) scores generated by the proposed method on optimizing QED and "QED+SA+JNK3+GSK3$\beta$". Then we compare our method with baseline methods on 3 molecules with the highest objective ($F$) scores in Table 5 and 6 for single-objective and multi-objective generation, respectively.

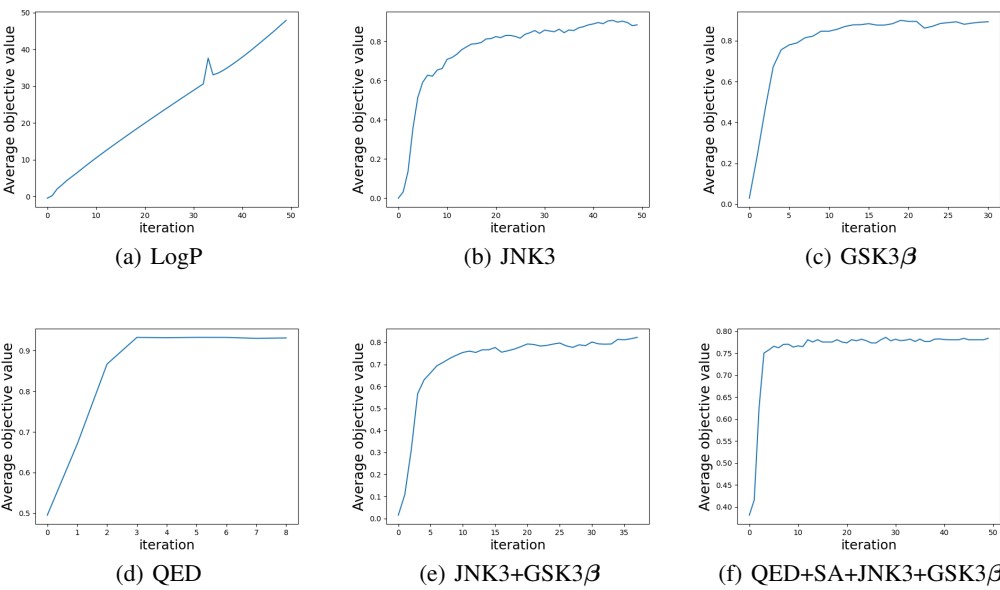

(a) LogP  (b) JNK3  (c) GSK3$\beta$

(d) QED  (e) JNK3+GSK3$\beta$  (f) QED+SA+JNK3+GSK3$\beta$

Figure 8: The optimization curves in *de novo* optimization experiments. The objective value ($F$) is a function of iterations.

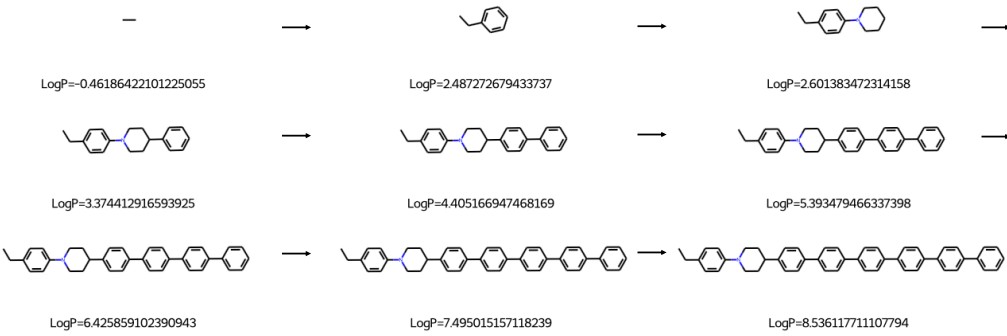

Figure 9: The first eight steps in *the de novo* optimization procedure of LogP. The model successfully learned to add a six-member ring to each step.

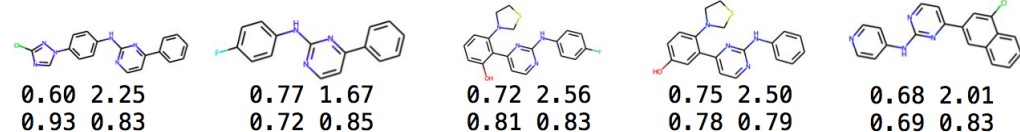

(a) Molecules with highest average QED, normalized-SA, JNK3 and GSK3$\beta$ scores, four scores represent QED, raw SA, JNK3, and GSK3$\beta$ scores, respectively.

(b) Molecules with highest QED.

Figure 10: Sampled molecules with the highest scores.

Table 5: Highest scores of generated molecules on single-objective *de novo* molecular generation. We present the result of `DST` in the first 50 iterations, but please note the setting of generation varies among the models, and an entirely fair comparison is impossible.

| Method | JNK3 | | | GSK3$\beta$ | | | LogP | | |
| --- | --- | --- | --- | --- | --- | --- | --- | --- | --- |
| | 1st | 2nd | 3rd | 1st | 2nd | 3rd | 1st | 2nd | 3rd |
| GCPN | 0.57 | 0.56 | 0.54 | 0.57 | 0.56 | 0.56 | 8.0 | 7.9 | 7.8 |
| MolDQN | 0.64 | 0.63 | 0.63 | 0.54 | 0.53 | 0.53 | 11.8 | 11.8 | 11.8 |
| GA+D | 0.81 | 0.80 | 0.80 | 0.79 | 0.79 | 0.78 | 20.5 | 20.4 | 20.2 |
| RationaleRL | 0.90 | 0.90 | 0.90 | 0.93 | 0.92 | 0.92 | - | - | - |
| MARS | 0.92 | 0.91 | 0.90 | **0.95** | 0.93 | 0.92 | 45.0 | 44.3 | 43.8 |
| DST | **0.97** | **0.97** | **0.97** | **0.95** | **0.95** | **0.95** | **49.1** | **49.1** | **49.1** |

## D.2 *De novo* MOLECULAR OPTIMIZATION ON QED (POTENTIAL LIMITATION OF DST)

As we have touched in Section 4.2, the optimization on QED is not as satisfactory as other oracles. We compare the performance of various methods on single-objective *de novo* molecular generation for optimizing QED score and show the result in Table 7. Additional baseline methods include JTVAE (junction tree variational autoencoder) (Jin et al., 2018) and GraphAF (Graph Flow-based Autoregressive Model) (Shi et al., 2020). The main reason behind this result is that our GNN predicts the target property based on a scaffolding tree instead of a molecular graph, as shown in Equation (7). A scaffolding tree omits rings' assembling information, as shown in Figure 2. Compared with other properties like LogP, JNK3, GSK3$\beta$, drug-likeness is more sensitive to *how* substructures connect (Bickerton et al., 2012). This behavior impedes the training of GNN and leads to the failure

Table 6: Highest scores of generated molecules on multi-objective *de novo* molecular generation. The score is the average value of all objectives.

| Method | JNK3+GSK3$\beta$ | | | QED+SA+JNK3+GSK3$\beta$ | | |
|---|---|---|---|---|---|---|
| | top-1 | top-2 | top-3 | top-1 | top-2 | top-3 |
| GCPN | 0.31 | 0.31 | 0.30 | 0.57 | 0.56 | 0.56 |
| MolDQN | 0.46 | 0.45 | 0.45 | 0.45 | 0.45 | 0.44 |
| GA+D | 0.68 | 0.68 | 0.67 | 0.71 | 0.70 | 0.70 |
| RationaleRL | 0.81 | 0.81 | 0.81 | 0.76 | 0.76 | 0.75 |
| MARS | 0.78 | 0.78 | 0.77 | 0.72 | 0.72 | 0.72 |
| DST | **0.89** | **0.89** | **0.89** | **0.83** | **0.83** | **0.83** |

of optimization. We report the learning curve in Figure 11, where we plot the normalized loss on the validation set as a function of epoch numbers when learning GNN. For fairness of comparison, validation loss is normalized by dividing the validation loss at scratch (i.e., 0-th epoch) so that all the validation losses are between 0 and 1. For most of the target properties, the normalized loss value on the validation set would decrease significantly, and GNN can learn these properties well, except QED. It verifies the failure of training the GNN on optimizing QED. A differentiable molecular graph at atom-wise resolution may potentially solve this problem.

Table 7: Comparison of different methods on optimizing QED for single-objective *de novo* molecular generation. The results for baseline methods are copied from You et al. (2018); Zhou et al. (2019); Shi et al. (2020); Xie et al. (2021). The results of JTVAE are copies from You et al. (2018).

| Method | top-1 | top-2 | top-3 |
|---|---|---|---|
| JTVAE (Jin et al., 2018) | 0.925 | 0.911 | 0.910 |
| GCPN (You et al., 2018) | **0.948** | 0.947 | 0.946 |
| MolDQN (Zhou et al., 2019) | **0.948** | **0.948** | **0.948** |
| GraphAF (Shi et al., 2020) | **0.948** | **0.948** | 0.947 |
| MARS (Xie et al., 2021) | **0.948** | **0.948** | **0.948** |
| DST | 0.947 | 0.946 | 0.946 |

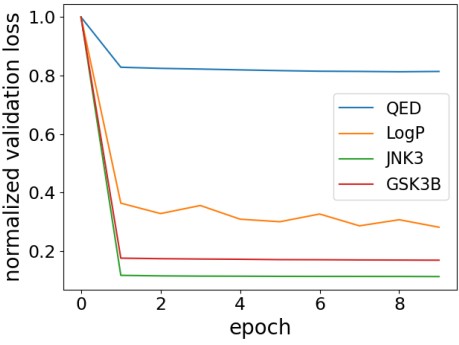

Figure 11: Normalized validation loss-epoch learning curves. For fairness of comparison, validation loss is normalized by dividing the validation loss at scratch (i.e., 0-th epoch) so that all the validation losses are between 0 and 1. For most of the target properties, the normalized loss value on the validation set would decrease significantly, and GNN can learn these properties well, except QED. The key reason for the failure of GNN on optimizing QED is the limitation of the expressive power of the scaffolding tree itself. QED is a property that is highly dependent on *how* substructures connect (Bickerton et al., 2012), while our scaffolding tree currently ignores that information. See Section D.2 for more details and analysis.

## D.3   RESULTS ANALYSIS FOR DISTRIBUTION LEARNING METHODS (LIGGPT)

As shown in Table 2 and 1, distribution learning methods (LigGPT) (Bagal et al., 2021) have much weaker optimization ability. DST and all the other baselines fall into the category of goal-directed

molecule generation, a.k.a., molecule optimization, which generates molecules with high scores for a given oracle. In contrast, LigGPT belongs to distribution learning (a different method category), which learns the distribution of the training set. We refer to Brown et al. (2019) for more description of two categories of methods. Consequently, conditioned generation learns from the training set, cannot generate molecules with property largely beyond the training set distribution, and can not optimize a property directly, even though they claim to solve the same problem. Problem formulation of distribution learning methods leads to an inability to generate molecules with property largely beyond the training set distribution, which means they are much weaker in optimization.

## D.4 ABLATION STUDY

As described in Section 3.3, during molecule sampling, we sample the new molecule from the differentiable scaffolding tree (Equation 11). To verify the effectiveness of our strategy, we compare with a *random-walk sampler*, where the topological edition (i.e., expand, shrink or unchanged) and substructure are both selected randomly. We consider the following variants:

- "DST + DPP". Both topology and substructure to fill are sampled from the optimized differentiable scaffolding tree, as shown in Equation (11). This is what we use in this paper.

- "random + DPP". Changing topology randomly, that is, at each leaf node, "expand", "shrink" and "unchange" probabilities are fixed to $0.5, 0.1, 0.4$. Substructure selection is sampled from the substructures' distribution in the optimized differentiable scaffolding tree. Then it uses DPP (Section 3.4) to select diverse and desirable molecules for the next iteration.

- "DST + top-$K$". Same as "DST + DPP", it uses DST to sample new molecules. The difference is when selecting molecules for the next iteration, it selects the top-$K$ molecules with highest $f$ score. It is equivalent to $\lambda \rightarrow +\infty$ in Equation (12).

We show the results in Figure 12. We find that both DST sampling and DPP-based diversification play a critical role in performance. We check the results for "DST + top-K", during some period, the objective does not grow, we find it is trapped into local minimum, impeding its performance, especially convergence efficiency. "random+DPP" exhibits the random-walk behavior and it would not reach satisfactory performance. When optimizing LogP, "DST +DPP" and "DST +top-$K$" achieved similar performance because logP score will prefer larger molecules with more carbon atoms, which is less sensitive to the diversity and relatively easier to optimize. Overall, "DST +DPP" is the best strategy compared with other variants.

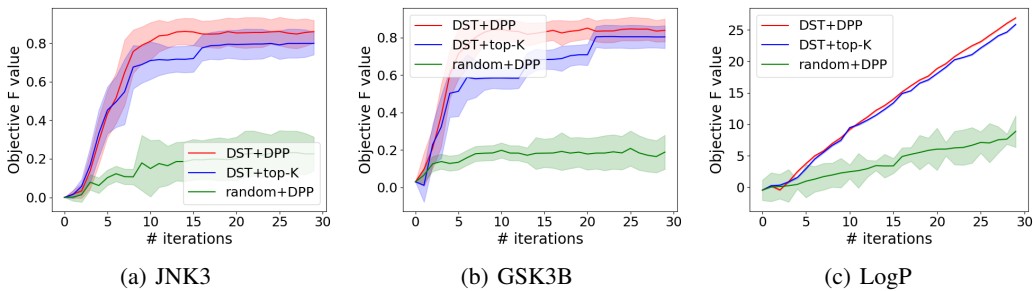

(a) JNK3          (b) GSK3B          (c) LogP

Figure 12: Ablation study. Objective value ($F$) as a function of iterations. See Section D.4 for more details.

## D.5 CHEMICAL SPACE VISUALIZATION

We use principal component analysis (PCA) to visualize the distribution of explored chemical structures in optimizing "JNK3 & GSK3$\beta$". Specifically, we fit a two-dimensional principal component analysis (PCA) (Bro & Smilde, 2014) for 2048-bit Morgan fingerprint vectors of 20K ZINC molecules, which are randomly selected from the ZINC database (Sterling & Irwin, 2015). Then we use the PCA to project the fingerprint of the generated molecule from various generations into a two-dimensional vector to observe their trajectories. The results are reported in Figure 13, where the grey points

represent the two-dimensional vector of ZINC molecules. We find that our method explores different parts of the 2D projection of the chemical space and covers a similar chemical space as the ZINC database after 20 to 30 iterations.

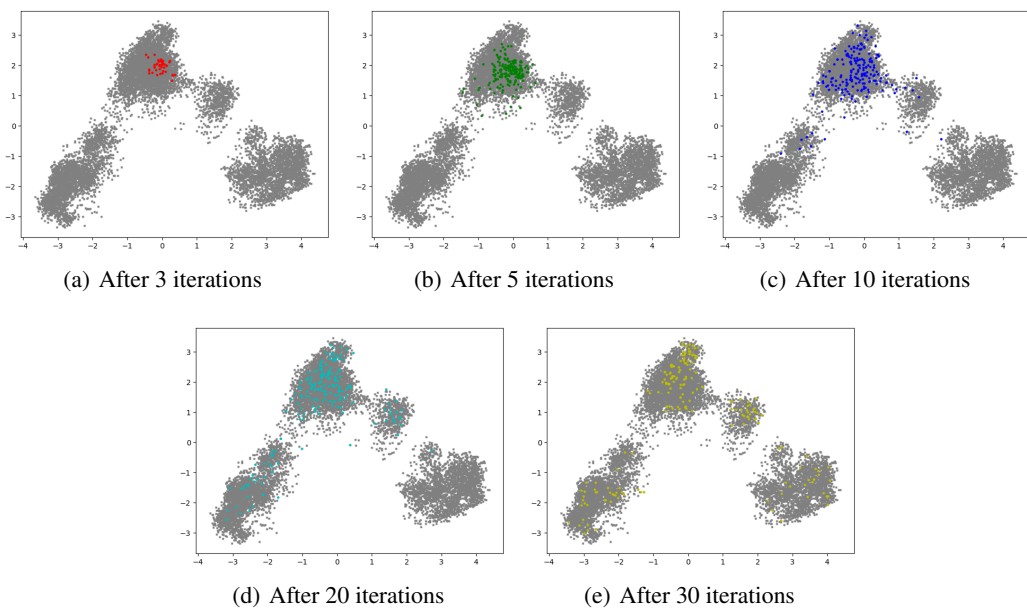

(a) After 3 iterations      (b) After 5 iterations      (c) After 10 iterations

(d) After 20 iterations      (e) After 30 iterations

Figure 13: Visualization of chemical space covered during optimization. We used PCA to reduce the dimension of Morgan's fingerprint. The gray points are the ZINC 250k data set. while colored points are generated molecules after corresponding iterations.

### D.6 ADDITIONAL INTERPRETABILITY ANALYSIS

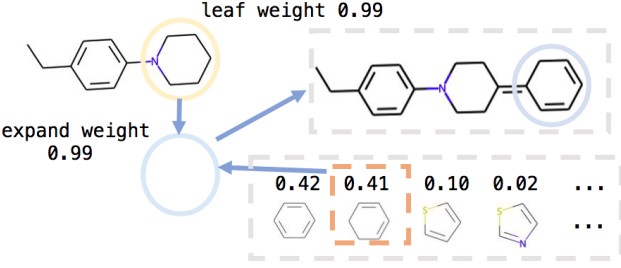

Figure 14: Interpretability analysis when optimizing LogP.

We provide an interpretability example in Figure 14. At the leaf node (yellow), from the optimized differentiable scaffolding tree, we find that the leaf weight and expand weight are both 0.99. Thus we decide to EXPAND, the six-member ring is selected and filled in the expansion node (blue). This is consistent with our intuition that the logP score will prefer larger molecules with more carbon atoms.

## E   THEORETICAL ANALYSIS

In this section, we present some theoretical results of the proposed method. First, in Section E.1, we conduct a convergence analysis of `DST` under certain mild assumptions.

### E.1 CONVERGENCE ANALYSIS

In this section, we discuss the theoretical properties of DST in the context of *de novo* molecule design (learning from scratch). We restrict our attention to a special variant of DST, named DST-greedy: at the $t$-th iteration, given one scaffolding tree $Z^{(t)}$, DST-greedy pick up only one molecule with highest objective value from $Z^{(t)}$'s neighborhood set $\mathcal{N}(Z^{(t)})$, i.e., $Z^{(t+1)} = \arg\max_{Z \in \mathcal{N}(Z^{(t)})} F(Z^{(t)})$ is exactly solved. We theoretically guarantee the quality of the solution produced by DST-greedy. First, we make some assumptions and explain why these assumptions hold.

**Assumption 1** (Molecule Size Bound). *The sizes (i.e., number of substructures) of all the scaffolding trees generated by DST are bound by $N_{min}$ and $N_{max}$.*

We focus on small molecule optimization; the target molecular properties would decrease significantly when the molecule size is too large, e.g., QED (drug-likeness) (Bickerton et al., 2012). Thus it is reasonable to bound the size of the scaffolding tree. In addition, we use submodularity and smoothness to characterize the geometry of the objective landscape.

**Assumption 2** (Submodularity and Smoothness). *Suppose $X_1, X_2, X_3$ are generated successively by DST-greedy via growing (i.e., EXPAND) a substructure on the corresponding scaffolding tree. We assume that the objective gain (i.e., $\Delta F$) brought by adding a single substructure would not increase as the molecule grows (EXPAND).*

$$F(X_3) - F(X_2) \leq F(X_2) - F(X_1), \quad \text{(submodularity)} \tag{14}$$

*where $X_2 = EXPAND(X_1, s_1)$, $X_3 = EXPAND(X_2, s_2)$, $s_1, s_2$ are substructures to add. Submodularity plays the role of concavity/convexity in the discrete regime. On the other hand, we specify the smoothness of the objective function $F$ by assuming*

$$F(X_3) - F(X_2) \geq \gamma(F(X_2) - F(X_1)), \ 0 < \gamma < 1 \quad \text{(smoothness)}$$

*holds for the $X_1, X_2, X_3$ described above, whose sizes are smaller than $N_{min}$.*

Then we theoretically guarantee the quality of the solution under these assumptions.

**Theorem 1.** *Suppose Assumption 1 and 2 hold; we have the following relative improvement bound with the optimum*

$$F(Z_*) - F(X_0) \geq \frac{1 - \gamma^{N_{min}}}{(1 - \gamma)N_{max}}\big(F(X_*) - F(X_0)\big), \tag{15}$$

*where $Z_*$ is the local optimum found by DST-greedy, $X_*$ is the ideal optimal molecule, $X_0$ is an empty molecule, starting point of de novo molecule design. In molecule generation setting, a molecule is a local optimum when its objective value is maximal within its neighbor molecule set, i.e., $F(Z_*) \geq F(Z)$ for $\forall Z \in \mathcal{N}(Z_*)$.*

The proof is given in Section G.4.

**Proof Sketch**. We first show that DST-greedy can converge to local optimum within a finite step in Lemma 1. Then we decompose the successive generation path and leverage the geometric information of the objective landscape to analyze the quality of the local optimum.

**Lemma 1** (Local optimum). *DST-greedy would converge to local optimum within finite steps.*

The proof is given in Section G.3.

## F EXTENSION OF MOLECULE DIVERSIFICATION

In the current iteration, we have generated $M$ molecules $(X_1, \cdots, X_M)$ and need to select $C$ molecules for the next iteration. We expect these molecules to have desirable chemical properties (high $F$ score) and simultaneously maintain higher structural diversity.

To quantify diversity, we resort to the *determinantal point process (DPP)* (Kulesza & Taskar, 2012). DPP models the repulsive correlation between data points (Kulesza & Taskar, 2012) and has been successfully applied to many applications such as text summarization (Cho et al., 2019), mini-batch

sampling (Zhang et al., 2017), and recommendation system (Chen et al., 2018). Generally, we have $M$ data points, whose indexes are $\{1, 2, \cdots, M\}$, $\mathbf{S} \in \mathbb{R}_+^{M \times M}$ denotes the similarity kernel matrix between these data points, $(i, j)$-th element of $\mathbf{S}$ measures the Tanimoto similarity between $i$-th and $j$-th molecules. We want to sample a subset (denoted $\mathcal{R}$) of $M$ data, $\mathcal{R}$ is a subset of $\{1, 2, \cdots, M\}$ with fixed size $C$, it assigns the probability

$$P(\mathcal{R}) \propto \det(\boldsymbol{S}_\mathcal{R}), \quad \text{where } \mathcal{R} \subseteq \{1, 2, \cdots, M\}, |\mathcal{R}| = C, \tag{16}$$

where $\boldsymbol{S}_\mathcal{R} \in \mathbb{R}^{C \times C}$ is the sub-matrix of $\mathbf{S}$, $\det(\boldsymbol{S}_\mathcal{R})$ is the determinant of the matrix $\boldsymbol{S}_\mathcal{R}$. For instance, if we want to sample a subset of size 2, i.e., $\mathcal{R} = \{i, j\}$, then we have $P(\mathcal{R}) \propto \det(\boldsymbol{S}_\mathcal{R}) = \boldsymbol{S}_{ii}\boldsymbol{S}_{jj} - \boldsymbol{S}_{ij}\boldsymbol{S}_{ji} = 1 - \boldsymbol{S}_{ij}\boldsymbol{S}_{ji}$, more similarity between $i$-th and $j$-th data points lower the probability of their co-occurrence. DPP thus naturally diversifies the selected subset. DPP can be calculated efficiently using the following method.

**Definition 11** (DPP-greedy (Chen et al., 2018)). *For any symmetric positive semidefinite (PSD) matrix $\boldsymbol{S} \in \mathbb{R}_+^{M \times M}$ and fixing the size of $\mathcal{R}$ to $C$, Problem* (16) *can be solved in a greedy manner by DPP-greedy in polynomial time $O(C^2 M)$. It is denoted $\mathcal{R} = DPP\text{-}greedy(\{X_1, \cdots, X_M\}, C)$.*

We describe the DPP-greedy algorithm in Algorithm 2 for completeness. Each iteration selects one data sample that maximizes the current objective, as described in Step 5 in Algorithm 2.

---

**Algorithm 2** DPP-greedy (Chen et al., 2018)

1: **Input**: symmetric positive semi-definite matrix $\mathbf{S} \in \mathbb{R}^{M \times M}$, number of selected data $C \in \mathbb{N}_+$, $C < M$.
2: **Output**: $\mathcal{R} \subseteq \{1, 2, \cdots, M\}, |\mathcal{R}| = C$.
3: $\mathcal{W} = \{1, 2, \cdots, M\}$.
4: **for** $i = 1, 2, \cdots, C$ **do**
5: $\quad j = \underset{k \in \mathcal{W}}{\arg\max} \ \log \det(S_{\mathcal{R} \cup \{k\}})$.
6: $\quad \mathcal{R} = \mathcal{R} \cup \{j\}$.
7: $\quad \mathcal{W} = \mathcal{W} - \{j\}$.
8: **end for**

---

As mentioned, our whole target is to select the molecules with desirable properties while maintaining the diversity between molecules. The objective is formulated as

$$\underset{\mathcal{R} \subseteq \{1, 2, \cdots, M\}, |\mathcal{R}| = C}{\arg\max} \mathcal{L}_{\text{DPP}}(\mathcal{R}) = \lambda \sum_{t \in \mathcal{R}} v_t + \log P(\boldsymbol{S}_\mathcal{R}) = \log \det(\boldsymbol{V}_\mathcal{R}) + \log \det(\boldsymbol{S}_\mathcal{R}), \tag{17}$$

where the hyperparamter $\lambda > 0$ balances the two terms, the diagonal matrix $\boldsymbol{V}$ is

$$\boldsymbol{V} = \text{diag}\big([\exp(\lambda v_1), \cdots, \exp(\lambda v_M)]\big), \quad \text{where } v_1 = F(X_1), \cdots, v_M = F(X_M), \tag{18}$$

where $v_i$ is the $F$-score of the $i$-th molecule (Eq. 1), $\boldsymbol{V}_\mathcal{R}$ is a sub-matrix of $\mathbf{V}$ indexed by $\mathcal{R}$. For any square matrix $\boldsymbol{M}_1, \boldsymbol{M}_2$ of the same shape, we have

$$\det(\boldsymbol{M}_1 \boldsymbol{M}_2) = \det(\boldsymbol{M}_1)\det(\boldsymbol{M}_2) = \det(\boldsymbol{M}_2)\det(\boldsymbol{M}_1),$$

we further transform $\mathcal{L}_{\text{DPP}}(\mathcal{R})$ as below to construct symmetric matrix,

$$\mathcal{L}_{\text{DPP}}(\mathcal{R}) = \log \det(\boldsymbol{V}_\mathcal{R}) + \log \det(\boldsymbol{S}_\mathcal{R}) = \log \det(\boldsymbol{V}_\mathcal{R}\boldsymbol{S}_\mathcal{R}) = \log \det\left(\boldsymbol{V}_\mathcal{R}^{\frac{1}{2}} \boldsymbol{S}_\mathcal{R} \boldsymbol{V}_\mathcal{R}^{\frac{1}{2}}\right), \tag{19}$$

where $\boldsymbol{V}^{\frac{1}{2}} = \text{diag}\big([\exp(\frac{\lambda v_1}{2}), \cdots, \exp(\frac{\lambda v_M}{2})]\big)$. Then we present the following lemma for the usage of the *DPP-greedy* method.

**Lemma 2.** *Suppose $\mathbf{S} \in \mathbb{R}^{M \times M}$ is the (Tanimoto) similarity kernal matrix of the $M$ molecules, i.e., $\boldsymbol{S}_{ij} = \frac{\mathbf{b}_i^\top \mathbf{b}_j}{\|\mathbf{b}_i\|_2 \|\mathbf{b}_j\|_2}$, $\mathbf{b}_i$ is the binary fingerprint vector for the $i$-th molecule, $V$ is diagonal matrix defined in Eq. (18), then we have (1) $\boldsymbol{V}^{\frac{1}{2}} \mathbf{S} \boldsymbol{V}^{\frac{1}{2}}$ is positive semidefinite; (2) $\boldsymbol{V}_\mathcal{R}^{\frac{1}{2}} \boldsymbol{S}_\mathcal{R} \boldsymbol{V}_\mathcal{R}^{\frac{1}{2}} = (\boldsymbol{V}^{\frac{1}{2}} \mathbf{S} \boldsymbol{V}^{\frac{1}{2}})_\mathcal{R}$.*

The proof is given in Section G.1.

Thus, Problem (19) can be transformed as

$$\arg\max_{\mathcal{R}\subseteq\{1,2,\cdots,M\},|\mathcal{R}|=C} \mathcal{L}_{\mathrm{DPP}}(\mathcal{R}) = \log\det\left(\left(\boldsymbol{V}^{\frac{1}{2}}\mathbf{S}\boldsymbol{V}^{\frac{1}{2}}\right)_{\mathcal{R}}\right), \qquad (20)$$

which means we can use *DPP-greedy* (Def. 11) to solve Problem (19) and obtain the optimal $\mathcal{R}$.

**Discussion**. In Eq. (17), we have two terms to specify the constraints on the molecular property and structural diversity, respectively. When we only consider the first term ($\lambda\sum_{t\in\mathcal{R}} v_t$), the selection strategy is to select $C$ molecules with the highest $F$ score for the next iteration, same as conventional evolutionary learning in Brown et al. (2019); Jensen (2019); Nigam et al. (2020).

On the other hand, if we only consider the second term in Eq. (17), we show the effect of selection strategies under certain approximations. Suppose we have $C$ molecules $X_1, X_2, \cdots, X_C$ with high diversity among them, then we leverage DST to optimize these $C$ molecules respectively, and obtain $C$ clusters of new molecules, i.e., $\hat{Z}_{11}, \cdots, \hat{Z}_{1l_1} \overset{\text{i.i.d.}}{\sim}$ DMG-Sampler($\widetilde{\mathbf{N}}^*_{(X_1)}, \widetilde{\mathbf{A}}^*_{(X_1)}, \widetilde{\mathbf{w}}^*_{(X_1)}$); $\cdots$ ; $\hat{Z}_{C1}, \cdots, \hat{Z}_{Cl_C} \overset{\text{i.i.d.}}{\sim}$ DMG-Sampler($\widetilde{\mathbf{N}}^*_{(X_C)}, \widetilde{\mathbf{A}}^*_{(X_C)}, \widetilde{\mathbf{w}}^*_{(X_C)}$). Then we present the following lemma to show that when only considering diversity, under certain assumptions, Problem (17) reduces to multiple chain MCMC methods.

In Eq. (17), $\lambda$ is a key hyperparamter, a larger $\lambda$ corresponds to more weights on objective function $F$ while smaller $\lambda$ specifies more diversity. When $\lambda$ goes to infinity, i.e., only considering the first term ($\lambda\sum_{t\in\mathcal{R}} v_t$), it is equivalent to selecting $C$ molecule candidates with the highest $F$ score for the next iteration, same as conventional evolutionary learning in Jensen (2019); Nigam et al. (2020).

On the other hand, if we only consider the second term, we show the effect of selection strategies under certain approximations. Suppose we have $C$ molecules $X_1, X_2, \cdots, X_C$ with high diversity among them, then we leverage DST to optimize these $C$ molecules respectively, and obtain $C$ clusters of new molecules, i.e.,

$$\hat{Z}_{11}, \cdots, \hat{Z}_{1l_1} \overset{\text{i.i.d.}}{\sim} \text{DST-Sampler}(\widetilde{\mathbf{N}}^*_{(X_1)}, \widetilde{\mathbf{A}}^*_{(X_1)}, \widetilde{\mathbf{w}}^*_{(X_1)});$$
$$\cdots ;$$
$$\hat{Z}_{C1}, \cdots, \hat{Z}_{Cl_C} \overset{\text{i.i.d.}}{\sim} \text{DST-Sampler}(\widetilde{\mathbf{N}}^*_{(X_C)}, \widetilde{\mathbf{A}}^*_{(X_C)}, \widetilde{\mathbf{w}}^*_{(X_C)})$$

Then we present the following lemma to show that when only considering diversity, under certain assumptions, Problem (17) reduces to multiple independent Markov chains.

**Lemma 3.** *Assume (1) the inter-cluster similarity is upper-bounded, i.e., $sim(\hat{Z}_{ip}, \hat{Z}_{jq}) \leq \epsilon_1$ for any $i \neq j$; (2) the intra-cluster similarity is lower-bounded, i.e., $sim(\hat{Z}_{ip}, \hat{Z}_{iq}) \geq 1 - \epsilon_2$ for any $i \in \{1, 2, \cdots, M\}$ and $p \neq q$; when both $\epsilon_1, \epsilon_2$ approach to $0_+$, the optimal solution to Problem (17) is*

$$\{\hat{Z}_{1p_1}, \hat{Z}_{2p_2}, \cdots, \hat{Z}_{Cp_C}\},$$

*where $p_c = \arg\max_p F(\hat{Z}_{cp})$ for $c = 1, \cdots, C$.*

The proof is given in Section G.2.

**Remark**. When the inter-cluster similarity is low enough, and intra-cluster similarity is high enough, our molecule selection strategy reduces to multiple independent Markov chains. However, these assumptions are usually too restrictive for small molecules.

# G  PROOF OF THEORETICAL RESULTS

In this Section, we provide the proof of all the theoretical results in Section E and F.

## G.1  PROOF OF LEMMA 2

*Proof.* (I) $\mathbf{V}^{\frac{1}{2}}\mathbf{S}\mathbf{V}^{\frac{1}{2}}$ is positive semidefinite.

First, let us prove similarity kernel matrix $\mathbf{S} \in RB^{M \times M}$ based on molecular Tanimoto similarity is positive semidefinite (PSD), we know that the $(i, j)$-th element of $\mathbf{S}$ measures the Tanimoto similarity between $i$-th and $j$-th molecules, i.e.,

$$\mathbf{S}_{ij} = \frac{\mathbf{b}_i^\top \mathbf{b}_j}{\|\mathbf{b}_i\|_2 \|\mathbf{b}_j\|_2},$$

where $\mathbf{b}_i \in [0, 1]^P$ is the $P$-bit fingerprint vector for the $i$-th molecule (in this paper, $P = 2048$). $\mathbf{S}$ can be decomposed as

$$\mathbf{S} = \mathbf{B}\mathbf{B}^\top,$$

where matrix $\mathbf{B}$ is the stack of all the normalized (divided by $l_2$ norm, $\| \cdot \|_2$) fingerprint vector, as

$$\mathbf{B} = \left[ \frac{\mathbf{b}_1}{\|\mathbf{b}_1\|_2}, \frac{\mathbf{b}_2}{\|\mathbf{b}_2\|_2}, \cdots, \frac{\mathbf{b}_P}{\|\mathbf{b}_P\|_2} \right] \in \mathbb{R}^{P \times M}.$$

For $\forall \, \mathbf{x} \in \mathbb{R}^M$, we have

$$\mathbf{x}^\top \mathbf{S}\mathbf{x} = \mathbf{x}^\top \mathbf{B}^\top \mathbf{B}\mathbf{x} = (\mathbf{B}\mathbf{x})^\top (\mathbf{B}\mathbf{x}) \geq 0.$$

Thus, $\mathbf{S}$ is PSD.

Then, similarly, for $\forall \, \mathbf{x} \in \mathbb{R}^M$, we have

$$\mathbf{x}^\top \mathbf{V}^{\frac{1}{2}} \mathbf{S} \mathbf{V}^{\frac{1}{2}} \mathbf{x} = \mathbf{x}^\top \left( \mathbf{V}^{\frac{1}{2}} \right)^\top \mathbf{B}^\top \mathbf{B} \mathbf{V}^{\frac{1}{2}} \mathbf{x} = (\mathbf{B}\mathbf{V}^{\frac{1}{2}}\mathbf{x})^\top (\mathbf{B}\mathbf{V}^{\frac{1}{2}}\mathbf{x}) \geq 0.$$

where $\mathbf{V}^{\frac{1}{2}}$ is diagonal matrix, so $\mathbf{V}^{\frac{1}{2}} = (\mathbf{V}^{\frac{1}{2}})^\top$. Thus, $\mathbf{V}^{\frac{1}{2}}\mathbf{S}\mathbf{V}^{\frac{1}{2}}$ is symmetric and positive semidefinite.

(II) $\mathbf{V}_{\mathcal{R}}^{\frac{1}{2}} \mathbf{S}_{\mathcal{R}} \mathbf{V}_{\mathcal{R}}^{\frac{1}{2}} = (\mathbf{V}^{\frac{1}{2}}\mathbf{S}\mathbf{V}^{\frac{1}{2}})_{\mathcal{R}}$.

Without loss of generalization, we assume $\mathcal{R} = \{t_1, \cdots, t_C\}$, where $t_1 < t_2 < \cdots, t_C$. $\mathbf{V}^{\frac{1}{2}}$ is diagonal.

$$\mathbf{V}_{\mathcal{R}}^{\frac{1}{2}} = \begin{bmatrix} \exp(\frac{\lambda v_{t_1}}{2}) & & \\ & \ddots & \\ & & \exp(\frac{\lambda v_{t_C}}{2}) \end{bmatrix},$$

where

$$v_{t_i} = F(X_{t_i})$$

is the objective function of $t_i$-th molecule $X_{t_i}$. The $i, j$-th element of $\mathbf{V}_{\mathcal{R}}^{\frac{1}{2}} \mathbf{S}_{\mathcal{R}} \mathbf{V}_{\mathcal{R}}^{\frac{1}{2}}$ is

$$\left( \mathbf{V}_{\mathcal{R}}^{\frac{1}{2}} \mathbf{S}_{\mathcal{R}} \mathbf{V}_{\mathcal{R}}^{\frac{1}{2}} \right)_{ij} = \exp\left(\frac{\lambda v_{t_i}}{2}\right) \mathbf{S}_{t_i t_j} \exp\left(\frac{\lambda v_{t_j}}{2}\right). \tag{21}$$

On the other hand, the $i, j$-th element of $\mathbf{V}^{\frac{1}{2}}\mathbf{S}\mathbf{V}^{\frac{1}{2}}$ is $\exp(\frac{\lambda v_i}{2})\mathbf{S}_{ij}\exp(\frac{\lambda v_j}{2})$. Then the $i, j$-th element of $\left( \mathbf{V}^{\frac{1}{2}}\mathbf{S}\mathbf{V}^{\frac{1}{2}} \right)_{\mathcal{R}}$ is

$$\left( (\mathbf{V}^{\frac{1}{2}}\mathbf{S}\mathbf{V}^{\frac{1}{2}})_{\mathcal{R}} \right)_{ij} = \exp\left(\frac{\lambda v_{t_i}}{2}\right) \mathbf{S}_{t_i t_j} \exp\left(\frac{\lambda v_{t_j}}{2}\right). \tag{22}$$

Combining Equation (21) and (22), we prove $\mathbf{V}_{\mathcal{R}}^{\frac{1}{2}} \mathbf{S}_{\mathcal{R}} \mathbf{V}_{\mathcal{R}}^{\frac{1}{2}} = (\mathbf{V}^{\frac{1}{2}}\mathbf{S}\mathbf{V}^{\frac{1}{2}})_{\mathcal{R}}$.

$\square$

### G.2 PROOF OF LEMMA 3

*Proof.* We consider two cases in the solution $\mathcal{R}$. (A) one molecule for each input molecule $Z_1, \cdots, Z_C$. (B) other cases. Our solution belongs to Case (A).

(A) First, we prove for (A), our solution is optimal. We consider the second term in Equation (17), $\mathbf{S}_{\mathcal{R}}$ is diagonal dominant. Also, a determinant function is a continuous function with regard to all

the elements. Thus, $det(\mathbf{S}_{\mathcal{R}}) = \prod_{i=1}^{C}(\mathbf{S}_{\mathcal{R}})_{ii}$ goes to 1. Intuitively, all the selected molecules are dissimilar to each other, and the diversity is maximized. On the other hand, to maximizing the first term in Equation (17), during each $k \in \{1, 2, \cdots, C\}$, we select molecule with highest $F$ score from $\{\hat{Z}_{k1}, \cdots, \hat{Z}_{kl_k}\}$. That is our solution.

(B) Then we prove all the possible combinations in (B) are worse than our solution. In (B), based on the pigeonhole principle, there is at least one input molecule $Z_k$ that corresponds to at least two selected molecules. Without loss of generalization, we denoted them $\hat{Z}_{k_1}$ and $\hat{Z}_{k_2}$. Since $\mathbf{S}_{\mathcal{R}}$ is diagonally dominant, its determinant can be decomposed as

$$\det(\mathbf{S}_{\mathcal{R}}) = \prod_{k=1}^{C} \det(\hat{S}_k).$$

If there is at least one $\hat{S}_k$ whose shape is greater than 1. Based on the definition of determinant, for matrix $A \in \mathbb{R}^{M \times M}$

$$\det(\mathbf{A}) = \sum_{\eta \in \text{Perm}(M)} \text{sgn}(\eta) \prod_{i=1}^{M} \mathbf{A}_{i,\eta(i)}, \tag{23}$$

where $\text{Perm}(M)$ is the set of all permutations of the set $\{1, 2, \cdots, M\}$, $\text{sgn}(\eta)$ denotes the signature of $\eta$, a value that is +1 whenever the reordering given by $\eta$ can be achieved by successively interchanging two entries an even number of times, and -1 whenever it can be achieved by an odd number of such interchanges. For exactly half of all $\eta$s, $\text{sgn}(\eta) = 1$ and the other half are equal to -1. For the matrix $A$ whose shape is greater than 1 and all the elements are equal to 1, the determinant is equal to $\sum_{\eta \in \text{Perm}(M)} \text{sgn}(\eta) = 0$.

The determinant function is a continuous function with regard to all the elements. When $\epsilon_2$ goes to $0_+$, all the elements of $\hat{S}_k$ approach to 1, the determinant goes to 0. Thus, $\det(\mathbf{S}_{\mathcal{R}})$ also goes to 0. The objective in Equation (17) goes to negative infinity. Thus, it is worse than our solution. $\square$

### G.3 PROOF OF LEMMA 1

*Proof.* For the *de novo* design, DST-greedy start from scratch (empty molecule). First, we show in this setting, there is no "REPLACE" or "DELETE" by mathematical induction and contradiction. Since we start from an empty molecule, at the 1-st step, the action is "EXPAND". Then we show the first $t$ steps are "EXPAND", the $(t+1)$-th step is still "EXPAND". Now we have $X^{(t)} = \text{EXPAND}(X^{(t-1)}, s_{t-1})$ (where $s_{t-1}$ is a substructure). Suppose the $(t+1)$-th step DST's action is "REPLACE", e.g., $X^{(t+1)} = \text{REPLACE}(X^{(t)}, s_t)$ (where $s_t$ is a substructure), based on definition of DST-greedy, we have $F(X^{(t+1)}) \geq F(X^{(t)})$. Since DST only REPLACE the leaf node, we find that $X^{(t+1)}$ and $X^{(t)}$ are both in neighbor molecule set of $X^{(t-1)}$, i.e., $\mathcal{N}(X^{(t-1)})$, which contradict with the fact that $X^{(t)} = \arg\max_{X \in \mathcal{N}(X^{(t-1)})} F(X^{(t-1)})$. Similarly, we show that there would not exist "DELETE".

Then based on Assummption 1, we find that DST-greedy converges at most $N_{\max}$ steps.

$\square$

### G.4 PROOF OF THEOREM 1

*Proof.* Based on the Proof of Lemma 1, we find that there is only "EXPAND" action, then we can decompose the generation path as follows. Starting from scratch, i.e., $X_0$, suppose the path to optimum $X_*$ is

$$X_0 \rightarrow X_1 \rightarrow X_2 \rightarrow \cdots \rightarrow X_{k_1} = X_*,$$

where each step, one substructure is added. The path produced by DST-greedy is

$$Z_0(X_0) \rightarrow Z_1 \rightarrow Z_2 \rightarrow \cdots \rightarrow Z_{k_2} = Z_*.$$

Based on the definition of each step, only one substructure is added.

Based on Assumption 1, we have $N_{\min} \leq k_1, k_2 \leq N_{\max}$. There might be some overlap within the first several steps, without loss of generalization, we assume $Z_k = X_k$ and $Z_{k+1} \neq X_{k+1}$, where $k$ can be $0, 1, \cdots, k_1$. Based on Assumption 2, we have

$$F(X_1) - F(X_0) \geq F(X_2) - F(X_1) \geq F(X_3) - F(X_2) \geq \cdots \geq F(X_{k_1}) - F(X_{k_1-1}).$$

Then, we have

$$k_1\big(F(X_1) - F(X_0)\big)$$
$$\geq \big(F(X_1) - F(X_0)\big) + \big(F(X_2) - F(X_1)\big) + \big(F(X_3) - F(X_2)\big) + \cdots + \big(F(X_{k_1}) - F(X_{k_1-1})\big)$$
$$= F(X_{k_1}) - F(X_0)$$

Thus, we get

$$F(X_1) - F(X_0) \geq \frac{1}{k_1}\big(F(X_{k_1}) - F(X_0)\big)$$

$$\geq \frac{1}{N_{\max}}\big(F(X_{k_1}) - F(X_0)\big) = \frac{1}{N_{\max}}\big(F(X_*) - F(X_0)\big). \tag{24}$$

Since $Z_0 = X_0$, according to the definition of greedy algorithm, we have $F(Z_1) \geq F(X_1)$. Based on Assumption 2, we have

$$F(Z_{N_{\min}}) - F(Z_{N_{\min}-1}) \geq \gamma\big(F(Z_{N_{\min}-1}) - F(Z_{N_{\min}-2})\big) \geq \gamma^2\big(F(Z_{N_{\min}-2}) - F(Z_{N_{\min}-3})\big)$$
$$\geq \cdots \geq \gamma^{N_{\min}-1}\big(F(Z_1) - F(Z_0)\big).$$

Based on Assumption 1, we have $F(Z_*) - F(Z_0) \geq F(Z_{N_{\min}}) - F(Z_0)$. Then we have

$$F(Z_*) - F(Z_0)$$
$$\geq F(Z_{N_{\min}}) - F(Z_0)$$
$$= \big(F(Z_{N_{\min}}) - F(Z_{N_{\min}-1})\big) + \big(F(Z_{N_{\min}-1}) - F(Z_{N_{\min}-2})\big) + \cdots + \big(F(Z_1) - F(Z_0)\big)$$
$$\geq (1 + \gamma + \gamma^2 + \cdots \gamma^{N_{\min}-1})\big(F(Z_1) - F(Z_0)\big) \tag{25}$$
$$= \frac{1 - \gamma^{N_{\min}}}{1 - \gamma}\big(F(Z_1) - F(Z_0)\big)$$

Combining Equation (24) and (25), we have

$$F(Z_*) - F(X_0) \geq \frac{1 - \gamma^{N_{\min}}}{(1 - \gamma)N_{\max}}\big(F(X_*) - F(X_0)\big).$$

We observe that objective $F$'s improvement is relatively lower bounded.

$\square$

