# OpenReview forum: "Differentiable Scaffolding Tree for Molecule Optimization"
_ICLR.cc/2022/Conference — ICLR 2022 Poster_

### Official Review · Reviewer_JXBo · 2021-11-02

**Correctness:** 4
**Technical Novelty And Significance:** 4
**Empirical Novelty And Significance:** 2
**Recommendation:** 6
**Confidence:** 5

**Main Review:**

The paper provides a few interesting ideas although it also contains some weaknesses.

Strengths:
1. **Differentiable scaffolding tree**: The DST is designed as a learnable graph input of molecules. By back propagating the parameters in A, N and w, it samples out new molecular structure along the direction of optimizing purposed properties. DST implies a novel direction of resolving the problem of lead optimization.
2. **Scaffold tree instead of molecular graph**: The proposed method smartly uses scaffold tree instead of molecular graph to alleviate invalid molecules.
3. **Determinantal point process**: The paper leverage DDP to keep expected properties as well as the diversity of generated molecules.

Weaknesses:
1. **restricted backbone**: The backbone of the oracle model (as GNN used in the paper) must be a differentiable model. This limits the usage of the proposed method. For example, if the oracle response comes from the real wet experiments, the proposed method is unable to get gradients from the wet experiments. Moreover, it may also suffer from the activity cliff of molecular properties.
2. **Local optimum**: SInce the proposed method generates new molecules step by step, it is possible that the selected pathway of generating new molecule is not an optimal solution. When it falls into a local optimum, the properties of the generated molecules will not be improved more.
3. **Costly assembling process from scaffold tree**: it is also a problem of JTVAE by practice. Assembling molecular graphs from scaffold tree is very time consuming.

Suggestions:
1. **Better assembling way of scaffold tree**: the authors could leverage some ideas from [1] to avoid costly assembling process.
2. **Add constraints to learnable N, A, and w**: Some lead optimization tasks may require un-changeable portions of the molecules. This could be easily achieved by adding constraints to learnable N, A, and w.

[1] Jin, Wengong, Regina Barzilay, and Tommi Jaakkola. "Hierarchical generation of molecular graphs using structural motifs." International Conference on Machine Learning. PMLR, 2020.

**Summary Of The Paper:**

The paper designs a novel model to optimize molecular leads with a differentiable scaffolding tree. The idea of learnable parameters in indicator matrix, adjacency matrix, and node weight vector is novel and interesting although the method still contains some weaknesses. The paper still provides a good direction of molecular generation.

**Summary Of The Review:**

Overall, it is a good paper with some interesting designs like DST. The proposed method shows a new direction to do lead optimization. I prefer to accept this paper although it contains some cons that hard to avoid.

---

> ### Author Response · Authors · 2021-11-20
> **Clarification of some issues**
>
> Thank you for your comments. Please find our response below. We also updated the draft and highlighted the modified part.
>
> COMMENT
> ---
>
> restricted backbone: The backbone of the oracle model (as GNN used in the paper) must be a differentiable model. This limits the usage of the proposed method. For example, if the oracle response comes from the real wet experiments, the proposed method is unable to get gradients from the wet experiments. Moreover, it may also suffer from the activity cliff of molecular properties.
>
> RESPONSE
> ---
>
> We are using the GNN as a surrogate to differentiate other properties. Most of the oracles we used in experiments (e.g., QED, SA) are not differentiable. Therefore, our approach can be used in combination with wet-lab experiments. The existence of activity cliffs makes the optimization problem more challenging in general, which could affect any optimization-based methods and is not unique to our method.
>
> -----------
>
> COMMENT
> ---
>
> Local optimum: Since the proposed method generates new molecules step by step, it is possible that the selected pathway of generating new molecule is not an optimal solution. When it falls into a local optimum, the properties of the generated molecules will not be improved more.
>
> RESPONSE
> ---
>
> The local optimum is a problem faced by all optimization algorithms except exhaustive search. We use Determinantal Point Process (DPP) to enhance the diversity of generated molecules to alleviate the issue. Specifically, instead of greedily choosing molecules with the highest properties, we diversify the generated molecule sets, i.e., minimizing the similarity between molecules so that it is less likely for DST to fall into a local optimum. This is validated to be helpful by the ablation study in Section D.4 and Figure 11.
>
> -----------
>
> COMMENT
> ---
>
> Costly assembling process from scaffold tree: it is also a problem of JTVAE by practice. Assembling molecular graphs from scaffold tree is very time consuming. Better assembling way of scaffold tree: the authors could leverage some ideas from [1] to avoid costly assembling process.
>
> RESPONSE
> ---
>
> We do not assemble molecular graphs from scaffolding trees from scratch at each step. Instead, within one iteration, we edit the given molecule using one operation (EXPAND, REPLACE or SHRINK), thus we only need to enumerate one branch in the scaffolding tree. The maximal generated candidates are 14, as shown in Fig 6b. We do agree that adopting more sophisticated building blocks could further improve, which is most likely application-specific.
>
> -----------
>
> COMMENT
> ---
>
> Add constraints to learnable N, A, and w: Some lead optimization tasks may require un-changeable portions of the molecules. This could be easily achieved by adding constraints to learnable N, A, and w.
>
> RESPONSE
> ---
>
> We agree. We can adapt to lead optimization tasks easily. Thanks for the recommendation.
>
> -----------
>
> Reference
> ===
>
> [1] Jin, Wengong, Regina Barzilay, and Tommi Jaakkola. "Hierarchical generation of molecular graphs using structural motifs." ICML, 2020.

---

> > ### Comment · Reviewer_JXBo · 2021-11-23
> > **Response to Authors**
> >
> > Thank you for your detailed response and additions to the paper. The authors' feedback addressed most of my concerns. Here I still have some concerns about the first comment.
> >
> > As the authors' response, the proposed method is able to deal with non-differentiable oracles. But from my understanding, in order to deal with such non-differentiable oracles, the proposed method has to pre-train a GNN model to learn the relationships between molecules and the query property. It actually builds an approximate function via DNN to approach the non-differentiable oracles. Therefore, it may be reasonable for QED and SA since the computational chemistry methods are reliable on those properties. But when the targeted properties are not computatable or not reliable by the computational chemistry, how to learn that approximate DNN? For example, the toxicity of the molecules is hard to compute and the data from wet lab is limited. The GNN trained from such small dataset is not reliable. Hence, the proposed method based on such GNN is not reliable, either. This may be an issue for the proposed method.
> >
> > But I might misunderstand here. If so, please let me know.
> >
> > Best,

---

> > > ### Author Response · Authors · 2021-11-23
> > > **Additional comments about oracles**
> > >
> > > Thanks for the additional comments. Yes, we need to first train a GNN surrogate for the oracle and we agree that directly optimizing an experimental oracle is challenging (to all de novo design methods, not only for ours), but it is possible. This is due to practical reasons, but not fundamental problem formulation.
> > >
> > > We first want to emphasize that the majority of the de novo molecular design algorithms (tree search, genetic algorithm, generative modeling, etc.), if not all, are only applied to optimize computational oracles. While a computational data point makes no difference from a data point measured in experiments from a machine learning perspective, one of the reasons that impedes the application of design algorithms to experimental oracle is that they require iteratively querying massive data during optimization (i.e., online data), which is too costly for most assays. As we showed in Table 1 and 2, the state-of-the-art algorithms still require tens or hundreds of thousands of data points (i.e., oracle calls) online to optimize an oracle. In contrast, ours require in total 15k-25k data points in those experiments, comprising mostly offline data that can be collected before optimization and only 5k online calls. This number of oracle calls is possible for some experimental oracle already. A real-world example of GNN applied to experimental data [1] shows it is possible to train a good enough GNN with 2k data points, and may require less if we adopt a pre-training strategy [2]. Hence, we think it is possible to optimize an experimental oracle with DST.
> > >
> > >
> > > References
> > > ===
> > >
> > > [1] Stokes, Jonathan M., et al. "A deep learning approach to antibiotic discovery." Cell 180.4 (2020): 688-702.
> > >
> > > [2] Hu, Weihua, et al. "Strategies for pre-training graph neural networks." ICLR 2020.

---

### Official Review · Reviewer_xvty · 2021-11-02

**Correctness:** 4
**Technical Novelty And Significance:** 4
**Empirical Novelty And Significance:** 4
**Recommendation:** 10
**Confidence:** 4

**Main Review:**

Difference to related works: Most of the existing deep generative models do not scale well on large datasets due to non-differentiable nature of molecule structures and they often heavily rely on brute-force enumeration. The proposed DST significantly increases scalability by employing implicit differentiation for gradient updates, i.e., fully differentiable generation of new molecules that preserves computation time and storage space without introducing any additional latency overhead or stochastic variation being induced by random initialization of weights in DST architecture.

Limitations: More quantitative performance comparisons with related works are needed.
The authors should expand on the approximation for formulating the problem as an iterative local discrete search specifically as it relates to equation 10.
Why did the authors allow for single atoms to be utilized as "substructures"? This aligns against the intuition of chemists who usually consider atoms as sidechains for modification as opposed to substructures for clustering or analyzing data by series.

**Summary Of The Paper:**

The molecular optimization is an essential chemical science and engineering task that has important applications such as drug discovery. Deep generative models and combinatorial optimization methods achieved initial success but still struggle with directly modeling discrete chemical structures and often heavily rely on brute-force enumeration. The challenge comes from the discrete and non-differentiable nature of molecule structures. To tackle this, the authors propose a differentiable scaffolding tree (DST) that utilizes a learned knowledge network to encode prior information.

Important contributions include:

(1) Definition of a local derivative of a chemical graph, we propose the differentiable scaffolding tree. The theory allows for gradient-based optimization of a discrete graph structure.

(2) The authors generalize a chemistry-aware graph network to encode prior knowledge. The network is capable of learning the hierarchy between functional groups and atoms, as well as the topological relations among chemical structure fragments.

(3) A unified graph-based optimization framework is proposed that can tackle both single-task and multi-task objective functions.

**Summary Of The Review:**

This paper created a molecular graph locally differentiable by using a differential scaffolding tree (DST). To the best of our knowledge, it does appear as a novel approach to make the molecular optimization problem differiable at the structure level rather than utilizing latent spaces or employing RL/evolutionary algorithms. The authors developed a generic molecular optimization approach based on DST, which was validated through comparison when previous methods and benchmark datasets. The paper appears well written and the results are promising.

---

> ### Author Response · Authors · 2021-11-20
> **More explanation about some setup and more experimental results.**
>
> Thank you for your positive comments. Please find our response below. We also updated the draft and highlighted the modified part.
>
> COMMENT
> ---
>
> More quantitative performance comparisons with related works are needed.
>
> RESPONSE
> ---
>
> In the updated paper, we added new baselines (ChemBO [1] and BOSS [2]) as suggested by “Reviewer BjiD”, both are Bayesian optimization methods. Also, we added the ablation study of a DST variant named **DST-rand** (random local search instead of optimizing and sample from DST), i.e., randomly selecting substructure from vocabulary and randomly selecting basic operations, including EXPAND, REPLACE, and SHRINK. The results are reported in Figure 3 and Table 1&2. We observe that DST outperformed these two BO methods (ChemBO and BOSS) and DST-rand in all the optimization tasks.
>
>
> -----------
>
> COMMENT
> ---
>
> The authors should expand on the approximation for formulating the problem as an iterative local discrete search specifically as it relates to equation 10.
>
> RESPONSE
> ---
>
> DST pipeline leverages local iterative search (Equation 9, Algorithm 1). In the t-th iteration, we optimize the DST of X^(t), i.e., X in Equation 10 is X^(t). We have added more explanations under Equation 10.
>
>
> -----------
>
> COMMENT
> ---
>
> Why did the authors allow for single atoms to be utilized as "substructures"? This aligns against the intuition of chemists who usually consider atoms as sidechains for modification as opposed to substructures for clustering or analyzing data by series.
>
> RESPONSE
> ---
>
> We agree that chemists typically consider atoms as sidechains attached to a scaffold. However, we are not simply using scaffolds for clustering analysis but are formulating a strategy to design novel molecules. We define “substructure” (basic building blocks, either a single ring or a single atom) to cover a large enough chemical space. In this case, we circumvent the difficulties of generating rings in molecular graphs and guarantee the 100% validity of generated molecules.
>
>
> References
> ===
>
> [1] Korovina, Ksenia, et al. "Chembo: Bayesian optimization of small organic molecules with synthesizable recommendations." International Conference on Artificial Intelligence and Statistics. PMLR, 2020.
>
> [2] Moss, Henry, et al. "BOSS: Bayesian Optimization over String Spaces." Advances in Neural Information Processing Systems 33 (2020).

---

### Official Review · Reviewer_5uXz · 2021-11-02

**Correctness:** 4
**Technical Novelty And Significance:** 3
**Empirical Novelty And Significance:** 3
**Recommendation:** 8
**Confidence:** 4

**Main Review:**

Pros:
- The idea proposed by authors seems novel and interesting.
- The results reported by authors are really promising.
- The proposed method allows to use a continuous gradient-based optimization of oracle function. Thus there is no need to train a latent space model before optimization.

Cons:
- In my opinion the section that describes the experimental settings  is not sufficiently developed. E.g. I cannot see how many molecules were generated and how many optimization iterations were performed for every molecule (the authors wrote that DST reached the highest scores within T = 50 iterations, but they do not specify whether they use this amount of steps in every experiments).

Questions during rebuttal period:
- How many adam steps were conducted in every iteration?

**Summary Of The Paper:**

The paper proposed differentiable scaffolding tree (DST), a novel method for molecular generation, that allows for a continuous gradient-based optimization of selected molecular properties. The results of de novo molecular optimization are really good. DST outperformed other methods in terms of optimization success rate, at the same time the proposed molecules were diverse and novel. The method also required less oracle calls (and majority of them before starting the optimization), compared to the other methods.

**Summary Of The Review:**

The method seems novel and interesting and results are promising. The only drawback is the weak section describing the experimental settings, which, however, can be easily corrected.

---

> ### Author Response · Authors · 2021-11-20
> **Clarification about some experimental setup**
>
> Thank you for your positive comments. Please find our response below. We also update the draft and highlight the modified part.
>
> -----------
>
> COMMENT
> ---
>
> In my opinion the section that describes the experimental settings is not sufficiently developed. E.g. I cannot see how many molecules were generated and how many optimization iterations were performed for every molecule (the authors wrote that DST reached the highest scores within T = 50 iterations, but they do not specify whether they use this amount of steps in every experiment).
>
> RESPONSE
> ---
>
> The number of distinct generated molecules is equal to the budget of calling an oracle, for a sample-efficiency comparison. Specifically, for all the multi- and single- objective optimization tasks (Table 1 and 2, optimizing “JNK3+GSK3beta”, “QED+SA+JNK3+GSK3beta”, “JNK3”, “GSK3beta”, “LogP”), DST generates 5K distinct molecules for each task, which costs 5K oracle calls and takes at most 50 iterations. We have clarified it in Section C.6 in the updated draft.
>
> -----------
>
> COMMENT
> ---
>
> How many Adam steps were conducted in every iteration?
>
> RESPONSE
> ---
>
> We conduct Adam steps 1000 times when optimizing DST in every iteration, enough to converge in almost every case. We have clarified it in Section C.6 in the updated draft.

---

### Official Review · Reviewer_BjiD · 2021-11-04

**Correctness:** 3
**Technical Novelty And Significance:** 4
**Empirical Novelty And Significance:** 2
**Recommendation:** 5
**Confidence:** 5

**Main Review:**

I thought that this paper was well-written and had a lot of genuinely interesting ideas. It was interesting to read. My main concern is that I think it is possible that the proposed algorithm is overly complicated and doesn't work as well as the authors claim. I will provide more details below.

== Pros ==

- To my knowledge, the differential scaffolding tree (DST) idea is novel. I think it is a very interesting contribution
- the details of the DST method are very well-described in section 3. I think it would be possible to re-implement their method based on the information given, which is a good sign
- Choice of experiments seemed good (I'm glad this is not yet another paper maximizing logP). It was nice to see that you included many baselines
- Good mentioning of relevant related work.
- The interpretability of the method is really interesting!

== Cons ==

- Looking at algorithm 1 (the main contribution), it is clearly a genetic algorithm. There is a population, and a loop over "generations" which includes a "mutation" step and a "new population selection" step. I found it slightly odd that the authors did not point out this connection, or use any terminology related to genetic algorithms. Given the similarity, it would be nice to see the authors discuss how their GA compares to previously proposed GAs for molecule optimization (such as the GAs from Guacamol, or the various SELFIES GAs by Alan Aspuru-Guzik's group)
- Thinking of the contribution as a genetic algorithm, it actually isn't clear to me that the whole DST thing is necessary. Fundamentally, the proposed mutation step takes a molecule X (in the form of a scaffold tree), and returns a set of candidate molecules whose scaffold tree differs from the current molecule's scaffold tree by 1 local editing operation (called $\mathcal{N}(X)$ in the paper). The DST effectively provides a way of sampling from $\mathcal{N}(X)$, where samples are (probably) predicted to have a high score by the GNN. While this is sensible, I can think of many similar alternatives which are much less complicated (see below). Overall I worry that the proposed method is just an overly complex way of doing something quite simple.
  1. Just enumerate all of $\mathcal{N}(X)$. My guess is that for small molecules this set is actually not that large ($\le 10,000$) although I could be wrong about this.
  2. Since the above approach may propose many poor molecules (and therefore not be sample efficient), one could enumerate $\mathcal{N}(X)$ and return a subset which the GNN predicts will have high scores. In fact, any oracle could be used here (e.g. random forest, GP) which might be less prone to overfitting and work with less training data.
  3. For computational efficiency, the approaches above could be modified to just randomly sample from $\mathcal{N}(X)$ rather than enumerating it. This is the approach taken by most GAs.
- Many details of the experimental section are not clear in the main text. As a reader I felt that the experiments offered little value without reading a lot of the appendix. Aware that there is a page limit, I nonetheless think that the following things needed to be in the main text:
  1. Brief explanation of the objectives. Why should I care about them? Are they novel objectives, or were they proposed in previous work? (seems like the latter)
  2. Why were your baselines chose, beyond just being other methods which work for molecular optimization?
  3. What is success rate? (turns out this is the only metric related to actually succeeding at optimization)
  4. Why should I care about novelty and diversity? Some readers might just care about finding $n$ molecules with high scores. I think there is a strong case for caring about these but it should at least be mentioned.
  5. How was the number of oracle calls chosen?
- Generally weak baselines w.r.t. sample efficiency: the authors claim that needing a small number of oracle calls is a key advantage of their method, and indeed, their method outperforms other methods with fewer oracle calls. Yet, as far as I can tell, none of the methods they compared with would actually be expected to succeed with a small number of oracle calls. GCPN, MolDQN, RationaleRL are all RL methods which only produce good molecules after a long training period. GA+D and MARS are exploratory methods which, as far as I understand, call the oracle indiscriminately on a lot of random points to explore molecular space. I am unfamiliar with LigGPT. To my knowledge, the most sensible baselines to compare with for sample efficiency would be Bayesian optimization. I suggest perhaps references [1-4] (in that order). If object that GPs don't scale and therefore are not applicable here, I would counter by saying that they will easily scale up to 10k points on a reasonable computer, and approximate GPs with 10k inducing points should be manageable with the 50k oracle calls shown in Figure 3.
- In table 1-2, the number of oracle calls is listed as if it is a fundamental property of each algorithm, which seems strange because it isn't: the number of oracle calls is directly decided by the number of iterations, the number of calls per iteration, etc, which can be chosen by the user. Simply put, all of the methods listed can be run with any number of oracle calls (although with fewer oracle calls performance will almost certainly be worse). It isn't clear whether the number of oracle calls was simply chosen so that DST had the lowest number, whether the number of oracle calls is really just the number of oracle calls necessary to meet some sort of external criterion (e.g. a certain number of successful molecules produced), or something else. In general, I think it is more fair to show comparisons like in Figure 3, which plots performance as a function of the number of oracle calls.
- Strange choice of metrics in Tables 1-2. I see many disadvantages to the metrics proposed in Tables 1-2. Generally, they are based around evaluating the entire set of molecules produced, rather than just evaluating the top points, which is more normal in optimization (since one generally cares about the optimum reached, not how one gets there). I particularly disagree with the use of "success rate" to measure optimization performance. Firstly, the cutoff for "success" seems fairly arbitrary. It is not clear how performance changes when this cutoff is changed. For this reason I think people usually report the top N scores or the average of the top N scores, rather than a measure of how many molecules exceed a threshold. Secondly, by measuring success as a "rate" the scores of sample inefficient methods seem to be unfairly penalized: even if they produce more molecules of better quality, the presence of many poor molecules (e.g. from the beginning of training in RL) will lower the score. Overall, I think that Figure 3 provides a much more valuable comparison of methods than Tables 1-2

== Suggestions for Improvement ==

Overall I think that the paper would benefit greatly from the following improvements:

1. Some sort of ablation study or comparison with modified GAs to determine whether the added complexity of the DST is actually necessary to achieve their optimization performance.
2. The addition of stronger baselines (I suggest specifically Bayesian optimization). This will help evaluate the performance of the proposed method.
3. Re-organization of the experimental section, including more explanations and reporting better metrics.

## References

[1] Korovina, Ksenia, et al. "Chembo: Bayesian optimization of small organic molecules with synthesizable recommendations." International Conference on Artificial Intelligence and Statistics. PMLR, 2020.

[2] Moss, Henry, et al. "BOSS: Bayesian Optimization over String Spaces." Advances in Neural Information Processing Systems 33 (2020).

[3] Moss, Henry B., and Ryan-Rhys Griffiths. "Gaussian process molecule property prediction with flowmo." arXiv preprint arXiv:2010.01118 (2020).

[4] Gómez-Bombarelli, Rafael, et al. "Automatic chemical design using a data-driven continuous representation of molecules." ACS central science 4.2 (2018): 268-276.

**Summary Of The Paper:**

Although the term "genetic algorithm" is not used in the paper, this paper effectively proposes a fancy genetic algorithm (GA) for molecule optimization. The contribution is chiefly in the mutation step of the GA, which roughly uses a pre-trained graph neural network to take the derivative with respect to the adjacency and node identity matrices of a graph to produce a probability distribution over mutated graphs. They show results on several different molecular optimization tasks which suggests that their algorithm outperforms several baselines.

**Summary Of The Review:**

I feel quite conflicted on this paper: on the one hand, the DST is to my knowledge novel and fairly interesting. On the other hand, the lack of "good" baselines makes it hard for me to judge the performance of this method, and I am concerned that a lot of the complexity of the DST is not actually necessary. Tentatively I think I will recommend rejection due to these concerns, but I would be happy to raise my score if they are addressed.

---

> ### Author Response · Authors · 2021-11-20
> **DST is not a genetic algorithm. We added more ablation studies, two Bayesian Optimization (BO) methods as baselines, and the requested metrics.**
>
> Thanks for your comments. Please find our response below. Modifications are highlighted in the draft.
>
> DST is not a genetic algorithm (GA)
> ---
>
> DST shares some similarities with GA. But strictly speaking, it is a variant of tree search but not GA because:
> - Tree search can **start from scratch** (e.g., a single node) and locally searches each branch of **intermediates independently (no interaction between intermediates)**, e.g.,[1].
> - GA **starts from a mating pool** (set of strings/graphs that serve as warm start) and requires **mutation and crossover between parents (interaction between intermediates, i.e., patents)** to obtain the next generation, e.g.,[2].
> - Our method can also **start from scratch** (a ring or an atom) and iteratively searches **multiple intermediates independently (no interaction between intermediates)** based on DST. It can be seen as an efficient way to traverse the tree.
>
> To clarify, we replace “population size” with “beam width” (number of intermediates we keep in each iteration) in Alg 1.
>
> Besides, we already included **GA+D**[3] (enhanced SELFIES GA, from Aspuru-Guzik group) in baselines. GA+D outperforms Graph GA in Guacamol & vanilla SELFIES GA significantly (Sec 4 in [3]). We find DST outperforms GA+D in all tasks in Table 1&2, Fig 3.
>
> --------
>
> Improvement 1: Ablation Study (enumeration or random sample)
> ---
>
> Enumeration is prohibitively expensive. Thus, we add a DST variant named **DST-rand** as you suggested. It randomly selects basic operations (EXPAND, REPLACE, SHRINK) and substructure from vocabulary, then selects subset of random samples with high (surrogate) GNN prediction scores. Fig 3 & Table 1-2 show DST obtains significant improvement over DST-rand in all tasks. Also, Sec D.4 shows molecule diversification (Sec 3.4) has a positive contribution to performance in three single-objective tasks.
>
> -------
>
> Improvement 2: Bayesian Optimization (BO) baseline
> ---
>
> We add ChemBO[4] & BOSS[5] as baselines. Results in Fig 3 & Table 1-2 show DST outperforms these two BO methods consistently and significantly in all optimization tasks.
>
> -------
>
> Improvement 3: new metrics & more explanations in experiments
> ---
>
> **new metrics** we select top-100 molecules with the highest property, then evaluate novelty, diversity and average score of top-100 molecules instead of success rate.
>
> **more explanation**
> - 1. **objectives** We follow existing works [3,4,7]. QED/LogP/SA measure drug-likeness/solubility/synthesizability. JNK3/GSK3B measure biological activity to the targets JNK3/GSK3B, related to Alzheimer’s disease [7].
> - 2. **baseline selection** They are all state-of-the-art methods, covering deep generative models, genetic algorithms, reinforcement learning, Bayesian optimization. GA, GCPN, MARS achieve competitive performances in “limited oracle call” setting [6].
> - 3. **success rate** We agree. It is replaced with average score of top-100 molecules.
> - 4. **why novelty & diversity** They are common metrics, following [3,4,7]. Also, current molecular design process, e.g., drug development, is typically hierarchical, comprising multiple rounds. One usually choose multiple computationally designed molecules to validate experimentally in the following round. Thus, a diverse batch covering multiple molecular scaffolds is more desirable. Novelty evaluates the ability to explore beyond a given library, which is also desirable in terms of patentability.
> - 5. **selecting oracle calls’ number** For each method in Table 1&2, we set the number of oracle calls so that property score nearly converges w.r.t. oracle call’s number. Iterations required to converge is an essential characteristic of an optimization method.
>
> --------
>
> COMMENT
> ---
>
> Oracle can be random forest (RF), GP, which are less prone to overfitting.
>
> RESPONSE
> ---
>
> We agree that the surrogate oracle model can be RF or GP, but it would impede flexibility of DST. In DST, GNN allows back-propagating gradient to graph structure (DST here) to edit scaffolding tree directly and guarantee 100% validity of the generated molecules. However, it does not hold for RF or GP.
>
> Reference
> ===
>
> [1] Luo et al, RASSE: a new method for structure-based drug design. J. Chem. Inf. Comput. Sci. 1996
>
> [2] Jensen, Jan H. A graph-based genetic algorithm and generative model/Monte Carlo tree search for exploration of chemical space. Chem. Sci. 2019
>
> [3] Akshat Kumar Nigam, Pascal Friederich, Mario Krenn, and Alán Aspuru-Guzik. Augmenting genetic algorithm with deep neural network for exploring chemical space. ICLR20
>
> [4] Korovina et al. ChemBO: Bayesian optimization of small organic molecule with synthesizable recommendation. AISTATS20
>
> [5] Moss et al. BOSS: Bayesian Optimization over String Space. NeurIPS20
>
> [6] Huang et al: Therapeutics Data Commons: Machine Learning Datasets & Tasks for Drug Discovery and Development. NeurIPS2021 Track on Datasets and Benchmarks
>
> [7] Jin et al. Multi-objective molecule generation using interpretable substructures. ICML20

---

> > ### Comment · Reviewer_BjiD · 2021-11-22
> > **Thank you for your response + further questions**
> >
> > Thank you very much for your detailed response. I really appreciated the changes that you have made to the manuscript. I have a few comments (which I don't expect a response to) and some questions which I would appreciate an answer to.
> >
> > ## Comments (no need to respond)
> >
> > 1. DST is a GA: without getting into a semantic argument about what a genetic algorithm is, I still think that DST is a genetic algorithm. The crux of your response, if I understood it correctly, is that DST is not a GA because it doesn't have interactions between parents (in GA lingo, it has no crossover, only mutation). However, from what I understand crossover is not a strictly necessary part of GAs: I've seen genetic algorithms that have only a mutation but no crossover step (i.e. no interaction between parents). My comment was intended as a tip to make your algorithm easier to understand for readers. _It is ok if what you propose is a genetic algorithm; it doesn't mean that it is "not novel" or anything._ I still personally believe that if DST were explained as a genetic algorithm it would be easier for the reader to see where the novelty/innovation of your work is. To be clear, this point isn't really affecting my score.
> > 2. "Improvement 3: new metrics & more explanations in experiments" Thank you for this, I am happy with the extra explanation and the changes that you made here.
> >
> > ## Questions (please respond)
> >
> > I would really appreciate answers to these questions, as they will help me decide how much to change my score by (i.e. if I want to vote to accept).
> >
> > 1. **Improvement 1: Ablation Study** Thank you for adding this, I think its a big step forward toward strengthening the paper. However, I couldn't find any details about how many random samples from the tree you took at each step. I presume that if you took only 10 that performance would be bad, whereas if you took ~1e6 then performance should likely be quite good. Can you please list all of the parameter values that you used for DST-rand?
> > 2. **Improvement 2: Bayesian Optimization (BO) baseline** Thank you for adding this. I was somewhat surprised to see that BO did not perform as well as I expected it to. I am thinking that this could be because the original implementations in these papers is not very good, or because the settings used in their paper didn't work very well for your problem. Can you please tell me:
> >     1. Did you try to tune any of the parameters of these algorithms (e.g. which acquisition function to use)?
> >     2. I saw that you had 50k oracle calls, but don't have any details about using approximate GPs. Can you please elaborate on how you performed GP inference on 50k data points?

---

> > > ### Author Response · Authors · 2021-11-23
> > > **Setups for baselines**
> > >
> > > Thank you for your additional comments. Please find our responses below. We have updated the modifications in the draft.
> > >
> > > Parameter values for DST-rand
> > > ---
> > >
> > > During each iteration, we randomly select operations (EXPAND,SHRINK, REPLACE) and substructure from vocabulary to generate at most $K_1$ molecules, select at most top-$K_2$ molecules with highest GNN surrogate predictions. Then we evaluate the $K_2$ candidates with the real oracle and select $C$ molecules as starting points for the next iteration. For fair comparison, $C$ is set to 10, same as DST. In order to get the near-optimal setup, we try different combinations of ($K_1$, $K_2$), and finally find that $K_1=1000$, $K_2=100$ achieves the best performance in terms of average of top-100 molecules’ score. Other setups of DST-rand follow DST. We add the details into Section B in the draft.
> > >
> > > -----------
> > >
> > > Did you try to tune any of the parameters of BO algorithms (e.g. which acquisition function to use)?
> > > ---
> > >
> > > We did tune the parameters for both ChemBO and BOSS to conduct a fair comparison:
> > >
> > > - Implementation details of ChemBO
> > >
> > >   - Regarding the kernel option, we leveraged the kernel proposed in ChemBO, which is the optimal-transport-based distance and kernel that accounts for graphical information explicitly. We did extra experiments to validate that it empirically outperforms the Tanimoto similarity kernel based on molecular fingerprint.
> > >
> > >   - As for the acquisition, we followed the original paper and adopt the ensemble method [1] using the EI (Expected Improvement), UCB (Upper Confidence Bound), and TTEI (Top-Two Expected Improvement) acquisitions, we conducted extra experiment and found it empirically outperforms a single acquisition in our scenario, which is consistent to the ChemBO paper.
> > >
> > > - Implementation details of BOSS
> > >
> > >   - BOSS leveraged Sub-sequence String Kernel (SSK) as kernel to support variable length inputs, and utilized genetic algorithm (GA) to optimize acquisition function efficiently under syntactical constraints. We empirically compared GA based acquisition optimizer with random search based acquisition optimizer (provided in the BOSS code repository) to validate the superiority of GA based optimizer.
> > >
> > >   - GA requires intensive oracle calls and is leveraged in the “inner-loop” of acquisition maximization, so we do not spend too much computational resources in a single GA process. In a single GA process, the population size is set to 100, the generation (evolution) number is set to 100. These setups are nearly optimal and tuning these hyperparameters didn’t improve the performance.
> > >
> > > We add the details into Section B in the draft.
> > >
> > > -----------
> > >
> > > Details about using approximate GPs. Can you elaborate on how you performed GP inference on 50k data points?
> > > ---
> > >
> > > GPs do not scale well with big data. To address this issue, we use **Subset of Data** [2], which uses a subset of all the data points to reduce the size of the kernel matrix and learn the surrogate model efficiently. Following [2], we selected a subset of data points using k-means clustering based on molecular fingerprint, and sampled data points from each cluster. The size of the subset is set to m=1000. The computationally complexity is O(m^3). We add the details into Section B in the draft.
> > >
> > > References
> > > ===
> > >
> > > [1] Kirthevasan Kandasamy et al. Tuning hyperparameters without grad students:
> > > Scalable and robust bayesian optimisation with dragonfly, JMLR 2020.
> > >
> > > [2] K. Chalupka et al. A framework for evaluating approximation methods for Gaussian process regression. JMLR 2013.

---

> > > > ### Comment · Reviewer_BjiD · 2021-11-23
> > > > **Thank you for the added details, I am satisfied with the responses.**
> > > >
> > > > Thank you for the details. I feel happy about the random search baseline now. I still think that the GP baselines could be improved (in particular the random subset method with K-means probably doesn't work well for high-dimensional sparse molecular fingerprints), but you have definitely made a solid effort. GP Bayesopt requires a lot of expertise to correctly tune in my experience, but the way you've done it should definitely yield a non-trivial baseline which your method is able to beat.
> > > >
> > > > Overall I feel convinced that your method now beats a set of non-trivial baselines.

---

> > > > > ### Author Response · Authors · 2021-11-24
> > > > > **Thanks for your comments**
> > > > >
> > > > > Thank you for the productive dialogue about our experiments! If there are any additional clarifications or experiments that would help convince you to raise your score above 5 (now that we have better established the non-triviality of our baselines), please let us know.

---

> > > > > > ### Comment · Reviewer_BjiD · 2021-11-28
> > > > > > **Additional question**
> > > > > >
> > > > > > I have some questions about your dataset:
> > > > > >
> > > > > > 1. Is this the dataset that you used? https://github.com/wengong-jin/icml18-jtnn/blob/master/data/zinc/train.txt
> > > > > > 2. how did you choose the 10k points to label for your DST method? Were they selected at random from the larger dataset? I can't find this detail in the paper.
> > > > > > 3. What are the top 100 scores in the training dataset for your objectives?

---

> > > > > > > ### Author Response · Authors · 2021-11-28
> > > > > > > **Responses to your additional questions**
> > > > > > >
> > > > > > > Thanks for the additional questions. Please find our responses below. We will add the revisions into the draft.
> > > > > > >
> > > > > > >
> > > > > > > QUESTION
> > > > > > > ---
> > > > > > >
> > > > > > > Is this the dataset that you used? https://github.com/wengong-jin/icml18-jtnn/blob/master/data/zinc/train.txt
> > > > > > >
> > > > > > > RESPONSE
> > > > > > > ---
> > > > > > >
> > > > > > > It is available at https://github.com/wengong-jin/icml18-jtnn/blob/master/data/zinc/all.txt (**all.txt** instead of **train.txt**). We will add it in Section 4.1 and Section C.1 in the revision.
> > > > > > >
> > > > > > > --------------
> > > > > > >
> > > > > > > QUESTION
> > > > > > > ---
> > > > > > >
> > > > > > > how did you choose the 10k points to label for your DST method? Were they selected at random from the larger dataset? I can't find this detail in the paper.
> > > > > > >
> > > > > > > RESPONSE
> > > > > > > ---
> > > > > > >
> > > > > > > Yes. They are randomly selected from ZINC 250K (link in above response). ZINC 250K contains around 250K druglike molecules extracted from the ZINC database [1]. We will add it in Section 4.1 and Section C.1 in the revision.
> > > > > > >
> > > > > > > ----------------
> > > > > > >
> > > > > > >
> > > > > > > QUESTION
> > > > > > > ---
> > > > > > >
> > > > > > > What are the top 100 scores in the training dataset for your objectives?
> > > > > > >
> > > > > > > RESPONSE
> > > > > > > ----
> > > > > > >
> > > > > > > We list the top 100 scores in the training dataset for all the objectives below. We will add in the Table 1&2 in the revision.
> > > > > > >
> > > > > > >
> > > > > > > | data | JNK3 | GSK3B | LogP | JNK3+GSK3B | QED+SA+JNK3+GSK3B |
> > > > > > > | --------------- | --------- | ----------- | ------------ | ----------- | ----------- |
> > > > > > > | Training data |  0.29  | 0.38 |  3.02 | 0.25 | 0.50 |
> > > > > > >
> > > > > > >
> > > > > > > REFERENCES
> > > > > > > ===
> > > > > > >
> > > > > > > [1] Teague Sterling and John J Irwin. Zinc 15–ligand discovery for everyone. Journal of chemical information and modeling, 55(11):2324–2337, 2015.

---

> > > > > > > > ### Comment · Reviewer_BjiD · 2021-11-29
> > > > > > > > **I tried a simple baseline and it seems to beat your method!?! *Response requested***
> > > > > > > >
> > > > > > > > Dear Authors,
> > > > > > > >
> > > > > > > > I am very, very sorry to be commenting so close to the end of the discussion period with an important point. Over the weekend I was thinking about whether to raise my score, and felt that I still had lingering doubts about the quality of the baselines. So, I decided to run the Graph genetic algorithm from Jensen et al (implemented in the Guacamol paper), with a small number of modifications. I ran this because I have found that the Graph GA method generally has strong performance with no tuning, and it is very fast to run. It ended up performing extremely well on the QED+SA+JNK3+GSK3β task, with the following results (in the format of Table 1 from your paper). In particular, note that the APS of this method is quite a bit higher than the one that you report for DST. Even though the diversity is lower, it is likely that one can find a diverse subset of molecules produced which still have good scores. I also had some promising preliminary results on the JNK3+GSK3 task which hasn't finished running yet, so I will not report it here.
> > > > > > > >
> > > > > > > > | Nov         | Div     | APS | # oracle |
> > > > > > > > |--------------|-----------|------------| --|
> > > > > > > > | 100%| 0.476     | **0.826**        | 2k+13k |
> > > > > > > >
> > > > > > > > If accurate, **this result makes me doubt that the empirical performance reported in your paper is as strong as you claim that it is.** As a genetic algorithm with no explicit model, one would not expect Graph GA to be very efficient, or to perform very well given a small number or oracle queries. If the Graph GA outperformed not only your method but also all the baseline methods, then perhaps the baselines were not strong enough, or not tuned correctly? Since I only ran this method one time, I also get the sense that if I did hyperparameter tuning of the Graph GA that performance could be even stronger. Since a lot of your paper's stated value is the high optimization performance of the DST method, this result makes the contribution seem less significant to me. Again, this only applies if my result is directly comparable to yours, which I ask about below.
> > > > > > > >
> > > > > > > > ### Description of the method
> > > > > > > >
> > > > > > > > The code I ran is based on Graph GA (from https://github.com/BenevolentAI/guacamol_baselines/tree/master/graph_ga). I made the following modifications:
> > > > > > > >
> > > > > > > > 1. Small change to the way the mating pool is formed, which gave a performance boost for my own research project.
> > > > > > > > 2. Tracking number of function evaluations in a dictionary
> > > > > > > > 3. Removing the limit on molecular size that was in the original codebase, allowing it to propose large molecules.
> > > > > > > > 4. Fix bug noted in the github issues where parent mol objects are modified during crossover, which isn't supposed to happen.
> > > > > > > >
> > > > > > > > I have anonymously uploaded the code and results as a zip file here: https://ufile.io/hbsz419w
> > > > > > > >
> > > > > > > > The results are in the "results" directory, which contains the top 100 molecules, diversity scores, and a list of all molecules queried with their scores. There is a `README` file which contains a command to reproduce the results. It takes less than 1h to run.
> > > > > > > >
> > > > > > > > ### Request from the authors: please confirm or correct my findings
> > > > > > > >
> > > > > > > > I know it is last minute, but I would appreciate it if you could validate my experimental result. I have tried to follow the methodology used in the paper, and took the objective functions from directly from the code that you uploaded. Although the dataset wasn't specified directly in the paper, you directed me to the correct dataset during the discussion, which I randomly subsampled to use in my experiment. My "best in dataset" numbers are quite similar to the ones you reported above. Please respond either saying:
> > > > > > > >
> > > > > > > > 1. Yes, my experimental methodology is correct, and my results are directly comparable with the ones that you report.
> > > > > > > > 2. No, there is some problem with my methodology such that the numbers are not directly comparable. If so, please point out what this is, and ideally how to fix it. If you could re-run my script to get updated numbers that would be appreciated. It should take <1h to run.
> > > > > > > >
> > > > > > > > Your response here will help immensely with the reviewer discussion. Thanks, and again very sorry that this is so last minute.

---

> > > > > > > > > ### Comment · Reviewer_JXBo · 2021-11-29
> > > > > > > > > **It is an interesting point**
> > > > > > > > >
> > > > > > > > > I agree with Reviewer BjiD's findings. So I will downgrade my score unless the author would provide a reasonable answer.
> > > > > > > > >
> > > > > > > > > Best,

---

> > > > > > > > > > ### Author Response · Authors · 2021-11-30
> > > > > > > > > > **Graph-GA does not beat DST. We use a more fair setup that does not sacrifice diversity.**
> > > > > > > > > >
> > > > > > > > > > Thanks for your additional effort to validate our model. But there is a clear misunderstanding of the task. You can’t sacrifice Diversity in these tasks as it will produce trivial variations of the same molecule, which is probably what happened to the results from graph-GA. Your result achieved .476 Diversity, while our DST reached 0.755 diversity in the task. Even many baselines achieved much higher diversity than .476. The claim of beating our method with a much lower diversity score is completely incorrect and misleading.
> > > > > > > > > >
> > > > > > > > > > More specifically, we use a very different setup that does not sacrifice diversity as follows.
> > > > > > > > > >
> > > > > > > > > > - Importance of diversity in drug discovery: Current molecular design process, e.g., drug development, is typically hierarchical, comprising multiple rounds. One usually chooses multiple computationally designed molecules to validate experimentally in the following round. Thus, a diverse batch covering multiple molecular scaffolds is more desirable.
> > > > > > > > > > - We use a different setup that does not sacrifice diversity. Specifically, for DST, as shown in **step 12 in Algorithm 1**, within each iteration, we select Phi from the set of all generated molecules Gamma, |Phi|=C (C=10) for the next iteration.
> > > > > > > > > >   - DST lets **Omega=Omega \cup Phi, and evaluate top-100 on Omega**.
> > > > > > > > > >   - The graph-GA you used lets **Omega=Omega \cup Gamma and evaluate top-100 on Omega**, which would increase APS but decrease diversity.
> > > > > > > > > > - One can easily produce low-diversity solutions of very high APS if we just do some perturbation around the top-1 molecule, which could happen to methods like Graph-GA.
> > > > > > > > > >
> > > > > > > > > > Finally, we would be happy to continue the conversation and perform more validation together with the reviewers. But we have to point out that the request of validating your code in a day is stressful and kind of unreasonable.

---

> > > > > > > > > > > ### Comment · Reviewer_BjiD · 2021-11-30
> > > > > > > > > > > **Performance vs diversity**
> > > > > > > > > > >
> > > > > > > > > > > Thank you for commenting on my results, I agree that asking you to do this at the last minute is very stressful and I'm very sorry that my comment came so close to the end of the review cycle. The short review cycle of these conferences is stressful for all of us :'(
> > > > > > > > > > >
> > > > > > > > > > > Given that you use a multiple evaluation criteria, I agree it was inappropriate for me to write that the Graph GA beats your method when it only beat it on one of the criteria. As you correctly pointed out, it is possible to produce a lot of similar molecules with high scores, and thereby achieve low diversity but a high APS. However, it was unclear whether this is all that the Graph GA did. Because you chose to report metrics on only the top 100 molecules, even if a diverse set of high scoring molecules was produced, if there is a small amount of non-diverse set of molecules with high scores the reported diversity will be very low.
> > > > > > > > > > >
> > > > > > > > > > > To examine this, I quickly tried to map out the Pareto frontier by choosing 100 diverse but high scoring molecules from the results `json` file that I shared yesterday, using a greedy method. The results are below. Let me emphasize that _these are **not** new results, they are just a re-calculation of the metrics by selecting a different 100 molecules, with the goal of mapping out Graph GA's pareto frontier_. I think it is fair to say that Graph GA "achieved" all these results.
> > > > > > > > > > >
> > > > > > > > > > > ```
> > > > > > > > > > > Div,APS
> > > > > > > > > > > 0.918,0.404     *4
> > > > > > > > > > > 0.896,0.464
> > > > > > > > > > > 0.860,0.524
> > > > > > > > > > > 0.771,0.637     *1
> > > > > > > > > > > 0.732,0.689     *2
> > > > > > > > > > > 0.705,0.722     *3
> > > > > > > > > > >
> > > > > > > > > > > 0.669,0.766
> > > > > > > > > > > 0.629,0.789
> > > > > > > > > > > 0.543,0.814
> > > > > > > > > > > 0.499,0.822
> > > > > > > > > > > ```
> > > > > > > > > > >
> > > > > > > > > > > Comparing with the results from Table 1, Graph GA is Pareto optimal (i.e. beats on all metrics) for LigGPT at `*4` above, GCPN/MolDQN/GA+D/ChemBO/BOSS at `*3`, RationaleRL/MARS at `*2`, and DST-rand at `*1`. At the very least, this suggests to me that these baselines were quite weak.
> > > > > > > > > > >
> > > > > > > > > > > Fortunately these Graph GA results were *not* Pareto optimal compared to the DST methods, and therefore your method is plausibly on the Pareto frontier (although since I only ran the Graph GA for 15k iterations it is possible that performance would be higher if I ran for 25k like you did). However, it is not very far from your method.
> > > > > > > > > > >
> > > > > > > > > > > ### My conclusions from this
> > > > > > > > > > >
> > > > > > > > > > > Thank you again for highlighting the importance of the diversity objective. If Graph GA was indeed a Pareto improvement on DST that would be very concerning. Thankfully my preliminary results do not show this. However, Graph GA can be seen as a Pareto improvement on all of your baselines, and this does make me think that the strength of the baseline methods could be improved.
> > > > > > > > > > >
> > > > > > > > > > > I suggest that in future versions of this manuscript (camera-ready if it is accepted here, or a future submission if not) that you either scalarize your objective, or explicitly map out the Pareto frontier of all methods to examine the trade-off between APS and diversity. Ultimately comparing multi-objective optimization methods is very difficult...

---

> > > > > > > > > > > > ### Comment · Reviewer_JXBo · 2021-11-30
> > > > > > > > > > > > **DST still outperforms graphGA**
> > > > > > > > > > > >
> > > > > > > > > > > > I downgrade my score because Reviewer BjiD showed that GraphGA could be better than the proposed method. But after the explanation from the author and the new results from Reviewer BjiD that DST still outperforms graphGA. Therefore, I would like to keep my original score.

---

### Comment · Area_Chair_tkjb · 2021-11-15
**Relationship between MolecularRNN/GCPN/GraphAF and the DST framework**

Dear Authors,

Thank you for your submission. I was wondering about the relationship between your work and MolecularRNN/GraphAF/GCPN and other related methods. All these methods iteratively build the molecule by sequentially adding atoms and bonds, modeling at each step a distribution.  They enable building what seems to be essentially a DST. More precisely, one could apply the developed by you GCN to the probabilistic outputs of GraphAF to optimize the distribution in each step. In this sense, GraphAF seems to induce or implement what is described as "[...] differentiable scaffolding tree to define a local derivative of a chemical graph. This concept enables a gradient-based optimization of a discrete graph structure.".

Could you please comment on this, and more broadly on the relationship and relative strengths of your proposed framework and MolecularRNN/GraphAF/GCPN (I am aware that GCPN was mentioned in the introduction)?

Thank you,

AC

---

> ### Author Response · Authors · 2021-11-20
> **We make the learning objective differentiable w.r.t molecule graph structure, while prior works make the learning objective differentiable w.r.t. NNs’ parameters. Our method directly optimizes molecular graph structure, while prior works indirectly search molecule graphs via optimizing NNs.**
>
> Thanks for your comments. In sum, we make the learning objective differentiable w.r.t. molecule graph structure, while prior works make the learning objective differentiable w.r.t. NNs’ parameters. Our approach directly optimizes molecular graph structure, while prior works indirectly search molecule graphs via optimizing NNs. We add **Figure 5** into draft to illustrate the main difference between DST and these existing methods.
>
>
> - These **existing methods** (including MolecularRNN/GraphAF/GCPN) construct new generative distributions over edge/node type with neural network (NN), i.e., given the intermediate as input, output the distribution of new nodes/edges. To generate molecules with a desired property, these methods adopt a reinforcement learning (RL) algorithm to guide the generation. Their **learning objective is differentiable w.r.t NNs’ parameters** and gradient descent is used to **optimize the NNs’ parameters**. However, generative modeling is inherently more challenging and RL is extremely data-hungry.
>
> - In contrast, **our model** utilized a supervised learned surrogate model (GNN) that takes scaffolding tree of the molecular graph, named **differentiable scaffolding tree (DST)**, as input and predict the property (e.g., QED, JNK3). DST specifies nodes’ & edges’ **distributions** implicitly. We back-propagate gradient from GNN to **optimize DST (nodes’ & edges’ distributions) directly**. That is, we make **predicted property differentiable w.r.t graph structure (i.e., DST)**.
>
> Bypassing the challenging generative modeling, our algorithm outperformed the three algorithms in terms of optimization ability and sample efficiency. Table 5 in GraphAF paper showed their results in optimizing LogP, and all of the three algorithms achieved significantly lower top-1 value than DST (DST: 49.1, GraphAF: 12.23, MolecularRNN: 8.63, GCPB: 7.98, the results of our model is in section D.1).

---

### Comment · Reviewer_BjiD · 2021-11-22
**Thank you to authors for highlighting changes in red**

This change makes it very easy to see what has been changed in the paper, thank you for doing this in your response :)

---

### Decision · Program_Chairs · 2022-01-20

**Decision:**

Accept (Poster)

**Comment:**

This paper was a tough call. The key contribution of the paper is a genuinely useful technique for generating chemical compounds satisfying desired properties. However, there are some key issues with paper.

Reviewer *BjiD* found out that baselines are weak. Most importantly, he run thorough experiments with GraphGA, outperforming by a significant margin the baselines. With minor tweaks (e.g. enabling generating larger molecules). GraphGA achieves comparable though slightly weaker results to DST. Importantly, as pointed out by Reviewer BjiD, there is an important flaw in the experiments that some methods have a cap on the number of atoms they can add. For example, on the logP optimization task, it is possible to optimize the score by just adding carbon atoms. I would like to thank very much Reviewer for going beyond and running these experiments.

All reviewers emphasized the novelty as a key contribution. In internal discussion, I raised concern about novelty and framing of the work. One could argue that any autoregressive model (i.e. adding atoms and bonds at each step) forms a DST. One could also argue that training LSTM to produce the distribution of interest, like in [1], is also a DST because the fine-tuned LSTM encodes the distribution of many molecules and is differentiable with respect to the distribution it encodes.

Despite these flaws, it is a solid contribution, which is likely to be useful for the community. Thank you for your submission, and it is my pleasure to recommend acceptance.

For the camera-ready please: (a) include a well-tuned GraphGA (implementing different tradeoffs of diversity and fitness), (b) include LSTM as implemented in Guacamole as baseline, (c) discuss much more clearly novelty of the work. Additionally please ensure that other baselines are not hampered by limit on number of atoms they can add.

References:

[1] Generating Focussed Molecule Libraries for Drug Discovery with Recurrent Neural Networks, Segler et al, [https://arxiv.org/abs/1701.01329](https://arxiv.org/abs/1701.01329)